# WaveDiffusion: Exploring Full Waveform Inversion via Joint Diffusion in the Latent Space

## Abstract

Full Waveform Inversion (FWI) is a vital technique for reconstructing high-resolution subsurface velocity maps from seismic waveform data, governed by partial differential equations (PDEs) that model wave propagation. Traditional machine learning approaches typically map seismic data to velocity maps by encoding seismic waveforms into latent embeddings and decoding them into velocity maps. In this paper, we introduce a novel framework that reframes FWI as a joint diffusion process in a shared latent space, bridging seismic waveform data and velocity maps. Our approach has two key components: first, we merge the bottlenecks of two separate autoencoders—one for seismic data and one for velocity maps—into a unified latent space using VQ to establish a shared codebook. Second, we train a diffusion model in this latent space, enabling the simultaneous generation of seismic and velocity map pairs by sampling and denoising the latent representations, followed by decoding each modality with its respective decoder. Remarkably, our jointly generated seismic-velocity pairs approximately satisfy the governing PDE without any additional constraint, offering a new geometric interpretation of FWI. The diffusion process learns to score the latent space according to its deviation from the PDE, with higher scores representing smaller deviations from the true solutions. By following this diffusion process, the model traces a path from random initialization to a valid solution of the governing PDE. Our experiments on the OpenFWI dataset demonstrate that the generated seismic and velocity map pairs not only exhibit high fidelity and diversity but also adhere to the physical constraints imposed by the governing PDE.

## 1 Introduction

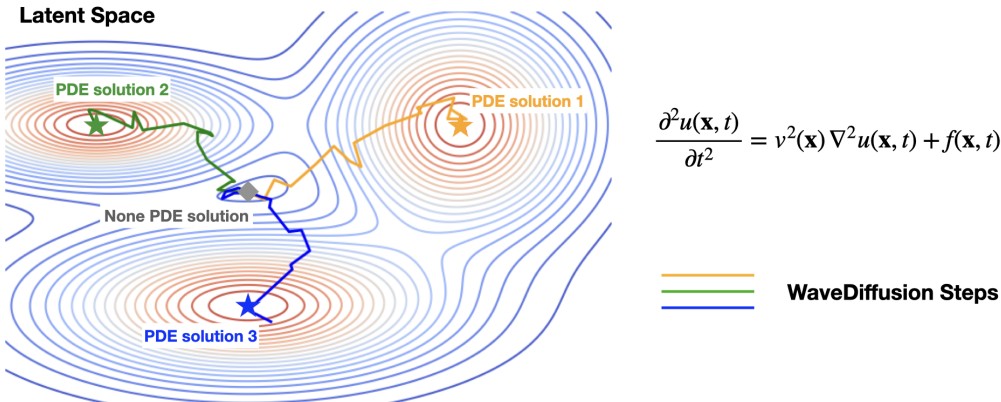

Figure 1: **Overview of WaveDiffusion**. WaveDiffusion refines the solution space of the governing wave equation by a diffusion process. Each peak (colored stars) corresponds to a PDE solution. The diffusion process transforms latent points in non-solution regions (gray squares) in the valleys toward accurate solutions consistent with the governing PDE.

Subsurface imaging is crucial for addressing many scientific and industrial challenges, such as understanding global earthquakes (Virieux et al., 2017; Tromp, 2020), monitoring greenhouse gas

storage (Li et al., 2021; Wang et al., 2023b), improving ultrasound medical imaging (Guasch et al., 2020; Lozenski et al., 2024), and aiding oil and gas exploration (Virieux & Operto, 2009; Wang & Alkhalifah, 2018). Subsurface structure is typically represented by acoustic wave velocity, which can be inferred from seismic data due to the physical relationship between them governed by a partial differential equation (PDE)—specifically, the acoustic wave equation (Sheriff & Geldart, 1995; Shearer, 2019). Full waveform inversion (FWI) is a powerful technique for reconstructing high-resolution subsurface acoustic velocity maps from seismic data. It is framed as a non-linear optimization problem: given seismic data $s$, the task is to solve for the velocity map $v$ according to the governing wave equation (Tarantola, 1984; Warner & Guasch, 2016).

Recently, machine learning-based approaches have been introduced to solve FWI (Wu & Lin, 2019; Sun & Demanet, 2020; Deng et al., 2022; Mousavi & Beroza, 2022). These methods typically use neural networks, particularly encoder-decoder architectures, to directly map seismic data to subsurface velocity maps, treating the FWI task as an image-to-image translation problem (Richardson, 2018; Feng et al., 2021; Jin et al., 2024). A more recent approach introduced generative diffusion models to regularize FWI by generating prior distributions for plausible velocity models, which guide the inversion process (Wang et al., 2023a). This approach treats FWI as a conditional generation problem, i.e. generating velocity maps using seismic data as condition.

In this paper, we explore a new direction by considering FWI as a *joint generative process*. In contrast to prior works that treat FWI as a conditional generation problem (where the velocity map is generated for a given seismic waveform), we are curious whether the two modalities—seismic waveform data and velocity map—can be generated simultaneously from a common latent space. Interestingly, we discovered that not only can these two modalities be jointly generated, but the generated seismic data and velocity maps also naturally satisfy the governing PDE without requiring any additional constraints. To achieve this, we propose a method with two key steps: first, we use a dual autoencoder architecture where both seismic data and velocity maps are encoded into a shared latent space, capturing the essential relationships between the two modalities. This shared latent space provides a coarse approximation of the wave equation solution space. Second, we apply a diffusion process in their common latent space, progressively refining the latent representations from random initializations. The corresponding decoders then generate seismic data and velocity maps from the refined latent representations. Our experiments on the OpenFWI dataset (Deng et al., 2022) empirically confirm that the jointly generated pairs satisfy the governing PDE.

This approach provides a new perspective on solving the governing wave equation. Unlike traditional FWI, which solves for one modality given the other, our method demonstrates that both seismic data and velocity maps can be solved simultaneously. This is achieved through a diffusion process within a shared latent space. Each point in this space corresponds to a seismic-velocity pair (after decoding), though not all points inherently satisfy the governing PDE. The diffusion model, however, learns to score each point based on its deviation from the PDE. By following the iterative denoising process, we trace a path from random initialization (with a lower score) to a valid solution (with a higher score) that satisfies both modalities of the PDE, as shown in Figure 1. With each denoising step, the deviation from the PDE decreases, leading to a more accurate solution.

It's important to note that our goal is not to push the boundaries of FWI performance but to offer a new perspective by extending FWI from a conditional generation problem to a joint generation problem. We hope that this approach will inspire deeper exploration and understanding within the research community, paving the way for new insights in computational imaging.

## 2 FWI OVERVIEW

Full Waveform Inversion (FWI) is a crucial technique for constructing high-resolution velocity maps of the subsurface from seismic data. The fundamental goal of FWI is to recover the velocity model $v(\mathbf{x})$ that governs the propagation of seismic waves, using observed seismic data $s(\mathbf{x_r}, t)$. In this work, seismic data refers to acoustic seismic data, which represents a simplified assumption of wave phenomena in the elastic Earth.

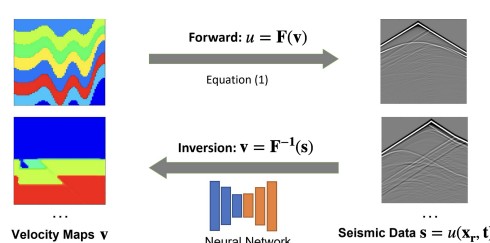

Figure 2: **Illustration of data-driven FWI:** forward simulation of seismic data $s$ via PDE $\mathbf{F}$ and inversion of $v$ via neural networks.

FWI is traditionally framed as an iterative optimization problem, where synthetic seismic data is generated through forward modeling and compared against observed data to update the velocity model iteratively. The synthetic data is typically computed by solving a partial differential equation (PDE) known as the wave equation. For the acoustic wave equation, this can be expressed as:

$$\frac{\partial^2 u(\mathbf{x}, t)}{\partial t^2} = v^2(\mathbf{x})\nabla^2 u(\mathbf{x}, t) + f(\mathbf{x_s}, t) \tag{1}$$

where $u(\mathbf{x}, t)$ represents the seismic wavefield, $v(\mathbf{x})$ is the subsurface velocity, $\nabla^2$ is the Laplacian operator, and $f(\mathbf{x_s}, t)$ is the source term. The objective of FWI is to minimize the misfit between observed seismic data $s_{\text{obs}}$ and synthetic data $s_{\text{syn}}$, generated by solving the wave equation. The typical data misfit objective function is:

$$\mathcal{L}_{\text{FWI}}(v) = \frac{1}{2}\sum_{t=1}^{T}\left\|s_{\text{obs}}(\mathbf{x_r}, t) - s_{\text{syn}}(\mathbf{x_r}, t; v)\right\|^2 \tag{2}$$

where $s_{\text{obs}}$ and $s_{\text{syn}}$ represent the observed and synthetic seismic data at receiver locations $\mathbf{x_r}$, respectively. The optimization process involves calculating the gradient of this loss function with respect to the velocity model using methods like the adjoint state method (Plessix, 2006). However, this iterative process is computationally expensive and often suffers from issues related to non-linearity and non-uniqueness in the inversion problem.

Recent advances in machine learning have introduced data-driven approaches for solving the FWI problem. These methods avoid the need for iterative PDE solvers by training neural networks to map seismic data directly to velocity maps. Approaches like *InversionNet* (Wu & Lin, 2019) treat the FWI problem as an image-to-image translation task, where convolutional encoder-decoder architectures are employed to generate velocity maps from seismic data in a single forward pass. This dramatically reduces computational costs compared to traditional methods.

However, while these machine learning-based models provide computational efficiency, they lack the physical grounding of traditional methods and often consider FWI as a conditional generation problem. As a result, they do not guarantee that the generated velocity maps satisfy the governing wave equation, potentially leading to physically inconsistent solutions.

## 3 WAVEDIFFUSION: OUR METHOD

In this section, we introduce WAVEDIFFUSION, an extension of FWI that moves beyond solving for velocity given seismic data, enabling the simultaneous solution of both modalities. Our approach involves two key steps: (1) a dual autoencoder architecture with vector quantization (VQ) that maps seismic data and velocity maps into a common latent space, and (2) a diffusion model applied in this common latent space, followed by two decoders that generate both modalities simultaneously. This method provides a new perspective on addressing the underlying PDE.

### 3.1 MOTIVATION FOR JOINT GENERATION

Existing machine learning-based approaches typically focus on translating one modality (seismic data) into another (velocity map) using encoder-decoder architectures. Mathematically, these methods provide only a partial solution to the governing PDE, as they rely on the availability of one modality. In contrast, our method tackles the more challenging problem: can both modalities be solved simultaneously?

Inspired by the success of diffusion models in generating multimodal outputs, such as images, audio, and video, we demonstrate that the joint distribution of seismic data and velocity maps can be modeled using a diffusion framework. This allows the two modalities to be generated simultaneously. Following the structure of latent diffusion models (Rombach et al., 2022), our approach involves two main steps: (1) a dual autoencoder with vector quantization and (2) a joint diffusion process for refining the latent representations.

### 3.2 PREPARING A SHARED LATENT SPACE WITH A DUAL AUTOENCODER

The WAVEDIFFUSION framework utilizes a dual autoencoder with vector quantization to construct a shared latent space as a preparatory step for the joint diffusion process. As illustrated in Figure 3,

the architecture features two encoder-decoder branches—one dedicated to seismic data and the other to velocity maps. Each encoder independently processes its input into a latent vector, which is then merged into a shared latent space $z$. This design builds upon existing approaches for multi-modal data representation (Feng et al., 2023; Chung et al., 2024).

Vector quantization discretizes the shared latent space, creating a compact and structured representation of the dependencies between seismic and velocity data. While this coarse latent representation does not yet adhere to the governing wave equation, it provides a foundation for refinement through the joint diffusion process, which is the central contribution of this work.

We emphasize that the dual autoencoder is not the focus of this framework and can be substituted with any architecture capable of producing a combined latent space for seismic data and velocity maps. For instance, the one-encoder-two-decoders architecture in Section 4.6 or an autoencoder incorporating KL divergence could also serve this purpose. The autoencoder primarily facilitates the setup for the joint diffusion process, which is the key innovation in achieving physically consistent seismic-velocity generation.

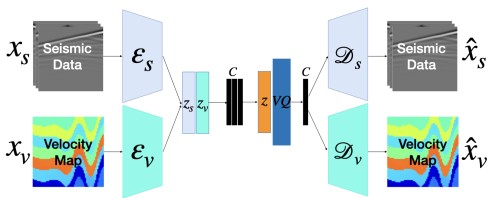

Figure 3: **Dual autoencoder architecture:** The autoencoder has two branches of encoders-decoders. The blue "VQ" block is the VQ layer. The orange block $z$ is the common latent space shared by seismic data and velocity maps, representing the coarse solution space of the PDE.

Figure 4: **Joint Diffusion architecture:** Gaussian noise is gradually added to the shared latent space $z$ for $t$ steps in the forward stage and progressively removed in the backward stage until the seismic-velocity pairs are consistent with the wave equation.

### 3.3 JOINT DIFFUSION PROCESS

**Latent Diffusion:** The joint diffusion process refines the coarse approximations produced by the autoencoder model by operating on the shared latent space $z$ of the two modalities. Gaussian noise is added during the forward process, progressively perturbing the latent vector. During the backward process, the noise is removed, guiding the model toward physically valid solutions (Figure 4). This iterative refinement ensures that the generated seismic-velocity pairs satisfy the wave equation.

1. **Forward Process:** Gaussian noise is added to the latent vector $z$, creating a noisy representation: $z_t = z + \epsilon_t$, where $\epsilon_t$ is the noise applied at step $t$.

2. **Backward Process:** The noisy latent vector is progressively denoised through $z_{t-1} = \mathcal{L}(z_t)$. Each backward step removes noise and refines the latent vector toward a valid solution that satisfies the wave equation.

**Sampling New Solutions:** After learning the backward denoising steps, new seismic-velocity pairs can be generated that satisfy the wave equation by sampling latent vectors from a Gaussian distribution and refining them using the learned backward steps.

1. Sample a latent vector $z_t$ from a standard Gaussian distribution: $z_t \sim \mathcal{N}(0, I)$.

2. Pass $z_t$ through the backward denoising steps: $z_{t-1} = \mathcal{L}(z_t), \quad z_{t-2} = \mathcal{L}(z_{t-1}), \dots, z_0$.

3. Decode $z_0$ back into seismic data and velocity maps: $\hat{x}_s = \mathcal{D}_s(z_0), \ \hat{x}_v = \mathcal{D}_v(z_0)$.

### 3.4 KEY INSIGHT: FROM COARSE TO FINE APPROXIMATION

**Coarse and Fine Approximation Discovery:** The autoencoder provides coarse approximations of the PDE solutions, producing seismic-velocity pairs that are plausible but do not fully satisfy

the wave equation (Figure 5 row 1). In contrast, the joint diffusion process refines these coarse approximations into physically valid solutions (Figure 5 row 2), demonstrating that the autoencoder captures general relationships between the modalities, while the diffusion refines them to satisfy the wave equation.

**Deviation Evaluation:** To quantitatively assess how well the generated seismic-velocity pairs adhere to the governing wave equation, we evaluate the deviation between the generated seismic data and the synthetic ground truth seismic data simulated using a finite difference (FD) solver using the jointly generated velocity map. Specifically, we analyze the L2 distance at each forward and backward diffusion step.

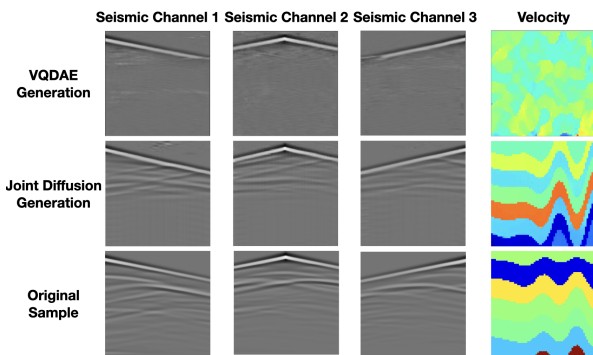

In diffusion, Gaussian noise is incrementally added to the latent vector $z$ during the forward process, and progressively denoised during the backward process. For each noisy latent vector $z_t$ at diffusion step $t$, we decode the seismic data $\hat{x}_\text{s}$ and velocity map $\hat{x}_\text{v}$ using the respective decoders.

Figure 5: **Comparison between generated and original samples:** Examples of generated pairs by the (row 1) autoencoder and (row 2) joint diffusion. We visually compare them to (row 3) an original OpenFWI example. The autoencoder-generated samples lack the physical relationship governed by the wave equation, while the joint diffusion generation refines them to better satisfy the PDE.

We then simulate synthetic seismic data $x_\text{s}(\hat{x}_\text{v})$ using a finite difference operator on the decoded velocity map $\hat{x}_\text{v}$. The L2 distance $\|\hat{x}_\text{s} - x_\text{s}(\hat{x}_\text{v})\|_2$ measures the deviation between the generated seismic data $\hat{x}_\text{s}$ and the FD-simulated synthetic seismic data $x_\text{s}(\hat{x}_\text{v})$. This evaluation is repeated for every forward and backward diffusion steps to track how the deviation changes as noise is added and removed.

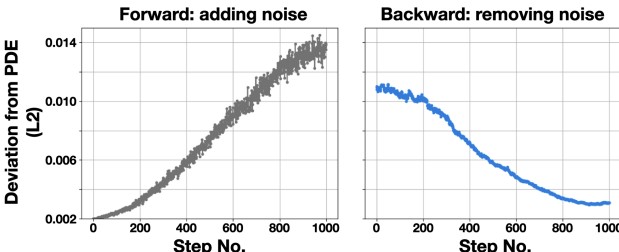

As noise is introduced during the forward process, the generated seismic data diverges from the physically valid solutions. Conversely, during the backward diffusion process, the model removes noise, progressively reducing the L2 distance and refining the seismic-velocity pairs toward solutions that better satisfy the wave equation. The statistical evaluation of 10000 generated pairs is shown in Figure 6, which illustrates how the deviation increases with noise in the forward process and decreases during

Figure 6: **Deviation from the governing PDE.** The L2 distance is calculated between (a) generated seismic data $\hat{x}_\text{s}$ and (b) synthetic seismic data $x_\text{s}(\hat{x}_\text{v})$ simulated from the generated velocity map using a finite difference solver.

denoising, confirming that the diffusion process scores the latent space based on how well the generated pairs adhere to the wave equation. Larger deviations from the PDE result in higher loss scores. During the backward diffusion process, these scores decrease as the generated pairs are refined into physically valid solutions (Figure 6 right).

**Transforming PDE into SDE:** The process of refining the latent space through diffusion can be interpreted as transforming the deterministic task of solving a PDE into a stochastic differential equation (SDE). As in Song et al. (2020), this SDE path allows exploration of a subspace of possible solutions, refining from coarse approximations to fine, physically consistent solutions.

## 4 EXPERIMENTS

In this section, we present the experimental evaluation of the proposed WAVEDIFFUSION framework. We conduct experiments on the OPENFWI dataset to evaluate the model's performance in

generating physically consistent seismic data and velocity maps. We assess the model's FID scores, analyze its ability to generate data that adheres to the governing PDE. Then we compare the results of training the state-of-the-art models such as *InversionNet* using the jointly generated dataset against the original OPENFWI benchmark. We further introduce an experiment to demonstrate how the joint diffusion model compares to separately trained diffusion models in generating seismic and velocity modalities. At last, we introduce how to use our joint diffusion model to perform a conventional FWI when only target seismic data is given.

## 4.1 DATASET AND TRAINING SETUP

We evaluate the performance of our WAVEDIFFUSION framework using the OPENFWI dataset, a benchmark collection of 10 subsets of realistic synthetic seismic data paired with subsurface velocity maps, specifically designed for FWI tasks. For our experiments, we focus on three representative subsets from the full dataset:

- **CurveVel_B (CVB)**: Subsurface structures with curved velocity layers.
- **FlatVel_B (FVB)**: Subsurface structures with flat velocity layers.
- **FlatFault_B (FFB)**: Flat layers intersected by faults.

We conducted two sets of training experiments:

1. **Individual subset training:** We trained the autoencoder and joint diffusion models separately on each of the three subsets (CVB, FVB, FFB) to evaluate their performance on individual geological structures.
2. **Combined dataset training:** We also trained the models on a combination of multiple datasets from OPENFWI to assess the model's generalization capability across a wider variety of geological structures.

Network details and training hyperparameters can be found in Appendix A.1.

## 4.2 EVALUATING AUTOENCODER RESULTS

The autoencoder model serves as the first stage of our WAVEDIFFUSION framework, producing coarse approximations of seismic data and velocity maps. When evaluated on the CVB subset, the autoencoder yielded an FID score of 14,207.14 for the generated velocity maps and 871.31 for the generated seismic data using an Inception-v3 model pre-trained on ImageNet (Szegedy et al., 2016). Similar results were observed for the other autoencoder models trained on the remaining subsets and the combined dataset. Visualization examples are shown in Figure 5 row 1. These high FID scores suggest that the autoencoder, while generating plausible structural shapes, does not adhere closely to the true data distribution. The large disparity between seismic and velocity FID scores indicates that the generated modalities deviate more from the physical relationships governed by the wave equation as coarse approximations of the PDE solutions.

## 4.3 JOINT DIFFUSION RESULTS AND ANALYSIS

The joint diffusion model is used to refine the coarse autoencoder generations into physically consistent seismic-velocity pairs. We evaluated the FID scores of both modalities generated by the joint

Table 1: **FID scores for various datasets.** The FID scores across the two modalities velocity maps $v$ and seismic data $s$ with different training datasets.

| Metrics \ Dataset | **CVB** | **FVB** | **FFB** | **3 sets** | **10 sets** |
|---|---|---|---|---|---|
| **Velocity FID** | 186.86 | 357.74 | 447.71 | 612.18 | 733.00 |
| **Seismic FID** | 30.66 | 88.01 | 34.05 | 74.58 | 128.64 |

diffusion model across the three individual subsets and combined datasets. As shown in Table 1, the best performance was achieved in the CVB subset, with an FID of 186.86 for velocity and 30.66 for seismic data. As the geological complexity increases (e.g., FFB), the FID scores rise, especially for the velocity modality, indicating the challenges posed by these more complex configurations.

We further evaluated the deviation from the governing PDE by tracking FID scores during the forward diffusion and backward denoising processes (Table 2). As noise is added in the forward pro-

cess, the FID scores increase significantly, peaking at 1000 timesteps (100% noise). On the other hand, as noise is progressively removed in the backward process, the FID scores decrease accordingly, reaching low values at the final step. This trend demonstrates how noise level is related to the deviation, and shows the model's ability to refine noisy representations into physically valid solutions through the denoising process.

Table 2: **FID score vs. noise level:** The FID scores for the two modalities along the forward and backward diffusion processes on the CVB dataset.

| Metrics \Process | Forward: adding noise | | | | Backward: removing noise | | | |
|---|---|---|---|---|---|---|---|---|
| Noise Level (%) | 0 | 20 | 50 | 100 | 100 | 50 | 20 | 0 |
| Velocity FID | 30.79 | 1135.82 | 3634.96 | 19393.46 | 19405.06 | 3569.01 | 620.33 | 186.86 |
| Seismic FID | 17.63 | 23.87 | 128.98 | 360.20 | 359.53 | 148.07 | 48.72 | 30.66 |

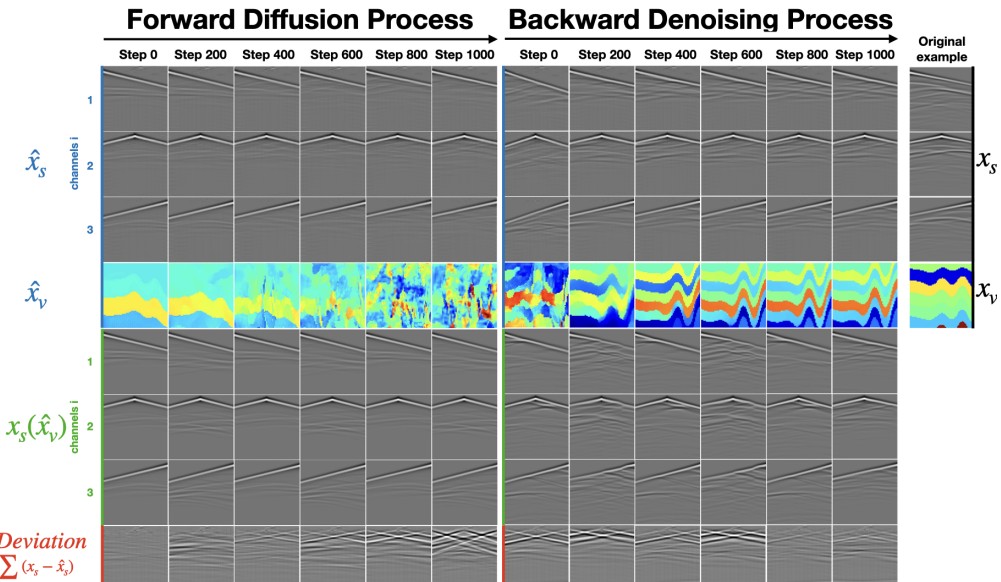

Figure 7: **Visualization of deviation from PDE during diffusion:** Seismic data comparison of the CVB example at different timesteps during the forward (left half) and backward diffusion processes (right half). The top three rows show seismic channels $\hat{x}_s$ generated by the joint diffusion model, while the fourth row shows the generated velocity map $\hat{x}_v$. Rows 5-7 show the synthetic seismic data $x_s(\hat{x}_v)$, simulated using a finite difference PDE solver on the generated $\hat{x}_v$. The last row shows the deviation from the governing PDE, visualized as the channel-stacked difference between $x_s(\hat{x}_v)$ and $\hat{x}_s$. Noise increases the deviation from the PDE during the forward diffusion, and the reverse process reduces discrepancies.

Figure 7 presents the results on the CVB dataset during the forward (adding noise) and backward (removing noise) diffusion processes. In the left half of Figure 7, during the forward diffusion process, as noise is added, the seismic data $\hat{x}_s$ generated by the joint diffusion model diverges more from the synthetic seismic data $x_s(\hat{x}_v)$, which is simulated using the generated velocity maps $\hat{x}_v$. This divergence, shown as the channel-stacked difference between $x_s(\hat{x}_v)$ and $\hat{x}_s$ in the last row, reflects the increasing deviation from the governing PDE as the noise level rises. In contrast, the right half of Figure 7 shows the backward denoising process, where noise is progressively removed, and $\hat{x}_s$ becomes more aligned with $x_s(\hat{x}_v)$, confirming the model's ability to refine the generated samples toward physically consistent solutions.

Similar trends can be observed in the separate FVB and FFB subset experiments and the combined datasets experiments, that in the forward diffusion, the generated modalities diverged from the ground truth, and the backward denoising converged back to PDE solutions. These results are included in the supplementary materials, further demonstrate the effectiveness of the joint diffusion model in generating realistic seismic data and velocity maps that adhere to the wave equation.

Results for the FVB and FFB single subsets can be found in Appendix A.2. Experiments with combined multiple datasets can be found in Appendix A.3.

## 4.4 COMPARISON WITH INVERSIONNET

Table 3: **Comparison of InversionNet performance on FFB dataset.** Performance of InversionNet trained with different datasets, comparing the jointly generated and original OPENFWI data.

| Setup | RMSE | MAE | SSIM |
|-------|------|-----|------|
| PureGen | 0.2623 | 0.1825 | 0.6484 |
| Gen+1% | 0.2396 | 0.1654 | 0.6614 |
| 1%Only | 0.3191 | 0.2436 | 0.5587 |
| OpenFWI | 0.1723 | 0.1106 | 0.7186 |

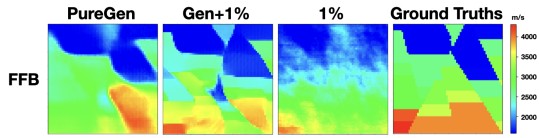

Figure 8: **InversionNet performance visualization.** The InversionNet predictions using the (column 1) WaveDiffusion generated samples, (column 2) WaveDiffusion generated plus 1% OpenFWI original samples, and (column 3) 1% OpenFWI original samples, compared to ground truth (column 4) on the FFB subset.

We evaluated the performance of InversionNet on the FFB dataset to assess how well the WAVEDIFFUSION-generated samples supplement the original OPENFWI dataset. The results in Table 3 compare the performance under three training setups:

**PureGen**: InversionNet trained on WAVEDIFFUSION-generated samples shows reasonable performance but falls short of the OPENFWI baseline, indicating a slight gap in the generated data's fidelity.

**Gen+1%**: Combining WAVEDIFFUSION-generated samples with 1% of the original OPENFWI dataset significantly improves performance, with metrics approaching the baseline. This highlights the effectiveness of augmenting generated data with even a small amount of real data.

**1%Only**: Training on only 1% of the original dataset results in the poorest performance, underscoring the inadequacy of such a small dataset for effective training.

These results demonstrate that while generated samples can effectively supplement small datasets, the inclusion of even a small portion of real data is crucial for achieving optimal results. Full results for other datasets (CVB and FVB) and additional comparisons are provided in the Appendix A.7.

## 4.5 SEPARATE VS. JOINT DIFFUSION

Table 4: **FID score comparison between separate and joint generations.** Evaluations on seismic data $s$ and velocity maps $v$ across CVB, FFB, and FVB datasets.

| Modality | Setup | CVB | FVB | FFB |
|----------|-------|-----|-----|-----|
| Velocity | Joint | 186.86 | 357.74 | 447.71 |
| | Separate | 411.40 | 360.90 | 385.32 |
| Seismic | Joint | 30.66 | 88.01 | 34.05 |
| | Separate | 131.48 | 179.69 | 117.63 |

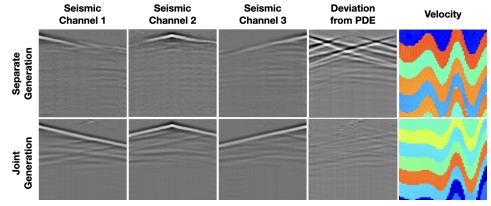

Figure 9: **Visualization of separate vs. joint diffusion.** Row 1: Separate models; Row 2: Joint model. Column 4 shows deviations from the governing PDE.

We compared the joint diffusion model with separate diffusion models, where seismic data and velocity maps were generated independently. In the separate models, the autoencoders lacked a shared latent space and used single-branch architectures. Both approaches were trained on the CVB, FFB, and FVB datasets. Table 4 and Figure 9 summarize the results.

The joint diffusion model consistently outperforms the separate models in FID scores across all datasets. For instance, on the CVB subset, joint diffusion achieves FID scores of 30.66 (seismic) and 186.86 (velocity), compared to 131.48 and 411.40, respectively, for the separate models. Similar trends are observed on FFB and FVB datasets, demonstrating the superior quality of the joint diffusion outputs.

Beyond visual quality, the joint model ensures physical consistency with the wave equation, which the separate models fail to achieve. Figure 9 highlights this distinction: the separate models show

significant deviations from the governing PDE, while the joint diffusion model generates seismic-velocity pairs that are both visually realistic and physically valid. This underscores the strength of the WAVEDIFFUSION framework in maintaining fidelity and physical correctness.

### 4.6 FWI OF TARGET SEISMIC DATA

This experiment demonstrates the capability of our joint diffusion model for FWI tasks where only seismic data is available, a realistic scenario in which direct access to velocity maps is infeasible.

We adapt the autoencoder to a one-in-two-out architecture, removing the velocity encoder and using only the seismic encoder. This ensures the latent space $z$ is fully derived from the seismic input. The latent vector $z$ is processed through two separate VQ layers and decoders to reconstruct both seismic data and velocity maps.

The joint diffusion phase refines $z$ to produce accurate inverted velocity maps, satisfying the physical constraints of the governing PDE while maintaining consistency with the seismic input. As shown in Figure 10, the model successfully reconstructs high-quality velocity maps (top row columns 2 & 4) and seismic data (bottom row columns 2 & 4) solely from seismic inputs (bottom row columns 1 & 3), even in the absence of target velocity maps (top row columns 1 & 3).

Table 5 compares our WAVEDIFFUSION model's performance on the CVB dataset with benchmarks from the OPENFWI paper. While VelocityGAN achieves slightly better RMSE, MAE, and SSIM scores, WAVEDIFFUSION performs competitively, showcasing its robustness and applicability for FWI tasks.

Furthermore, the model demonstrates resilience to noisy input seismic data, producing high-quality, noise-free inverted velocity maps. This denoising capability, discussed in Appendix A.5, highlights the versatility of WAVEDIFFUSION for practical FWI applications, even under challenging data conditions.

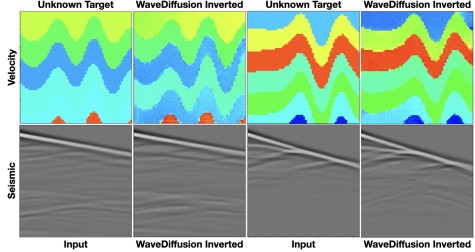

Table 5: **Performance comparison of FWI between WAVEDIFFUSION and OPEN-FWI.** RMSE, MAE, and SSIM are reported for seismic-to-velocity inversion on the CVB dataset.

| Network | RMSE | MAE | SSIM |
|---|---|---|---|
| WaveDiffusion (ours) | 0.2958 | 0.1824 | 0.6290 |
| InversionNet | 0.2801 | 0.1497 | 0.6727 |
| VelocityGAN | **0.2611** | **0.1268** | **0.7111** |
| UPFWI | 0.3179 | 0.1777 | 0.6614 |

Figure 10: **FWI Results of Target Seismic Data.** Top row: target velocity (column 1, 3) and inverted velocity (column 2, 4, respectively). Bottom row: input seismic data (column 1, 3) and inverted seismic data (column 2, 4, respectively).

## 5 RELATED WORKS

In this section, we review three major approaches relevant to our work: traditional physics-based FWI methods, machine-learning-based FWI approaches, and the use of generative models in FWI.

### 5.1 TRADITIONAL PHYSICS-BASED FWI

Traditional FWI methods aim to reconstruct subsurface velocity models by iteratively minimizing the difference between observed and simulated seismic data, typically using gradient-based optimization methods. The key challenge lies in solving the wave equation, which governs wave propagation through the Earth. While effective, these methods are computationally expensive and sensitive to factors such as the quality of the initial velocity model, noise in the data, and cycle-skipping issues—where the inversion algorithm converges to incorrect solutions due to poor starting models or insufficient low-frequency data (Tarantola, 1984; Virieux & Operto, 2009). Techniques such as adaptive waveform inversion (Warner & Guasch, 2016) and multiscale FWI (Bunks et al., 1995) have been developed to reduce the risk of cycle-skipping and improve convergence by progressively introducing higher-frequency data. These techniques remain the treatment of FWI as a conditional generation problem using physical equations as the engine (Virieux et al., 2017; Warner & Guasch, 2016).

## 5.2 Data-Driven Approaches to FWI

In recent years, machine learning approaches have been increasingly explored for FWI. Convolutional Neural Networks (CNNs) have shown promise in learning image-to-image mappings from seismic data to velocity models, bypassing the need for iterative solvers. Encoder-decoder architectures, such as those used in *InversionNet* (Wu & Lin, 2019) and *VelocityGAN* (Zhang et al., 2019), have demonstrated the ability to predict velocity maps from seismic data while reduces computational costs by learning implicit relationships between the two modalities. Richardson's work (Richardson, 2018) further illustrated that deep learning models could predict velocity models efficiently. However, these approaches are still treating the FWI problem as an image-to-image translation task or a conditional generation problem (Zhu et al., 2019; Wang et al., 2023a). Recent work on neural operators (Li et al., 2020; 2023) offers a more flexible approach by learning operators that map between the two modalities in a revered direction, i.e. predicting seismic data given velocity maps. Though these neural operator methods have shown powerful capacity in mapping the two modalities, they are still constrained by their image-to-image mappings setup and cannot solve for multiple modalities simultaneously that satisfy the governing PDE.

## 5.3 Generative Models in FWI

Generative models, particularly Generative Adversarial Networks (GANs) and their variants, have emerged as alternatives to traditional CNN-based methods for FWI. These models aim to learn the latent representations of seismic data and velocity models, enabling the generation of synthetic training data or even direct inversion (Goodfellow et al., 2020). Vector Quantized GANs (VQ-GANs) (Esser et al., 2021), in particular, have been explored for their ability to generate high-quality modalities, such as images, audios, videos, etc. Such models can be tuned for imaging one physical modality (e.g. velocity map) given another (e.g. seismic data) (Zhang et al., 2019).

Recent work has focused on Latent Diffusion Models (LDMs) (Ho et al., 2020; Dhariwal & Nichol, 2021; Rombach et al., 2022), which refine latent space representations through a diffusion process. LDMs iteratively denoise latent variables, progressively improving the quality of generated samples. While these models can produce realistic-looking data, they often generate new samples of one single modality at a time. Thus, it is difficult for them to generate multiple modalities using one generative model as they lack the physical consistency to the governing PDEs that describe the relationship between these modalities. Diffusion models have been applied to FWI by Wang et al. (Wang et al., 2023a), who used them to generate prior distributions for plausible velocity models as a regularization term. Their method still treats seismic data and velocity maps separately, limiting its ability to generate physically consistent seismic-velocity pairs.

## 6 Conclusion

In this paper, we introduced WAVEDIFFUSION, a novel framework that redefines FWI as a *joint generative process* for seismic-velocity pairs within a shared latent space. Unlike existing approaches that either solve for one modality given the other or treat FWI as a conditional generation problem, our method simultaneously solves for both modalities. This is achieved through a joint diffusion process on a shared latent space, established via a dual autoencoder, which progressively refines latent representations to satisfy the governing PDE. By tracing a path from random initializations to valid PDE solutions, WAVEDIFFUSION offers a new geometric perspective on the FWI problem.

Experiments on the OpenFWI dataset validate the effectiveness of WAVEDIFFUSION, demonstrating its ability to generate high-quality seismic-velocity pairs that exhibit strong fidelity and adhere to the physical constraints of the wave equation. Moreover, our framework can be directly applied to conventional FWI tasks by using a one-in-two-out configuration, where only seismic data is input to the encoder. Additionally, training an existing FWI model, such as InversionNet, on WAVEDIFFUSION-generated samples improves performance, particularly in scenarios with limited real data availability. These results highlight the versatility and practicality of WAVEDIFFUSION in advancing FWI research.

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

# A  APPENDIX

## A.1  NETWORK DETAILS AND TRAINING HYPERPARAMETERS

In this appendix section, we provide details on the network architectures and training hyperparameters used for the autoencoder and joint diffusion models in our experiments.

### A.1.1  AUTOENCODER MODEL

The autoencoder uses separate convolutional encoder-decoder branches that are constructed by ResNet blocks (He et al., 2016) for seismic and velocity data. The channel multipliers of the ResNet blocks are set to [1, 2, 2, 4, 4] for velocity maps and [1, 2, 2, 4, 4, 4, 4, 8, 8] for seismic data. The resolution for velocity maps is 64, while for seismic data it is [1024, 64]. The latent dimension $z$ is [16, 16], and the number of residual blocks is set to 3. The model was trained with a base learning rate of $4.5 \times 10^{-4}$. It uses an embedding dimension of 32 and an embedding codebook size of 8192. The autoencoder employs a perceptual loss combined with a discriminator. The discriminator starts training at step 50001 with a discriminator weight of 0.5 and a perceptual weight of 0.5.

### A.1.2  JOINT DIFFUSION MODEL

The Joint Diffusion model is based on the `LatentDiffusion` architecture. The backbone network in the Joint Diffusion model is a UNet-based architecture. The UNet takes 32 input and output channels, and the model channels are set to 128. The attention resolutions are [1, 2, 4, 4], corresponding to spatial resolutions of 32, 16, 8, and 4. The model uses 2 residual blocks and channel multipliers of [1, 2, 2, 4, 4]. It also employs 8 attention heads with scale-shift normalization enabled and residual blocks that support upsampling and downsampling. The model is trained with a base learning rate of $5.0 \times 10^{-5}$ and uses 1000 diffusion timesteps. The loss function applied is $L_1$. The diffusion process is configured with a linear noise schedule, starting from 0.0015 and ending at 0.0155.

A LambdaLinearScheduler is used to control the learning rate, with 10000 warmup steps. The initial learning rate is set to $1.0 \times 10^{-6}$, which increases to a maximum of 1.0 over the course of training.

### A.1.3  TRAINING HYPERPARAMETERS

Both the autoencoder and Joint Diffusion models were trained using the Adam optimizer, with $\beta_1 = 0.9$ and $\beta_2 = 0.999$. The models were trained with a batch size of 256 for 1000 epochs. The learning rate follows an exponential decay schedule with a decay rate of 0.98. Gradient clipping was applied with a threshold of 1.0. Early stopping was implemented when the validation loss plateaued for 10 consecutive epochs.

Training required approximately 1000 GPU hours for the autoencoder (per dataset) and 2000 GPU hours for the joint diffusion model.

Seismic data and velocity models were resized from [5,70,1000]/[1,70,70] to [3,64,1024]/[1,64,64] (channel, height, depth) for consistency with our architecture. Both were normalized to [-1,1] to ensure compatibility and stability.

## A.2  FVB AND FFB RESULTS

In addition to the CVB dataset, we conducted similar experiments on the FVB and FFB subsets. These subsets were chosen to evaluate the joint diffusion model's performance in more straightforward geological scenarios—flat velocity layers and flat layers with faults.

The results from both FVB and FFB mirrored the trends observed with CVB, as shown in Figures 11 and 12, respectively. Specifically, during the forward diffusion process, the seismic data generated by the joint diffusion model increasingly diverged from the finite difference simulation data as noise was introduced. Conversely, during the reverse denoising process, the generated seismic data converged towards the "ground truth" seismic data. This consistent behavior across different geological settings further validates the effectiveness of the joint diffusion in refining the latent space to satisfy

the wave equation constraints. Detailed results and visualizations for the FVB and FFB subsets are provided in the supplementary materials.

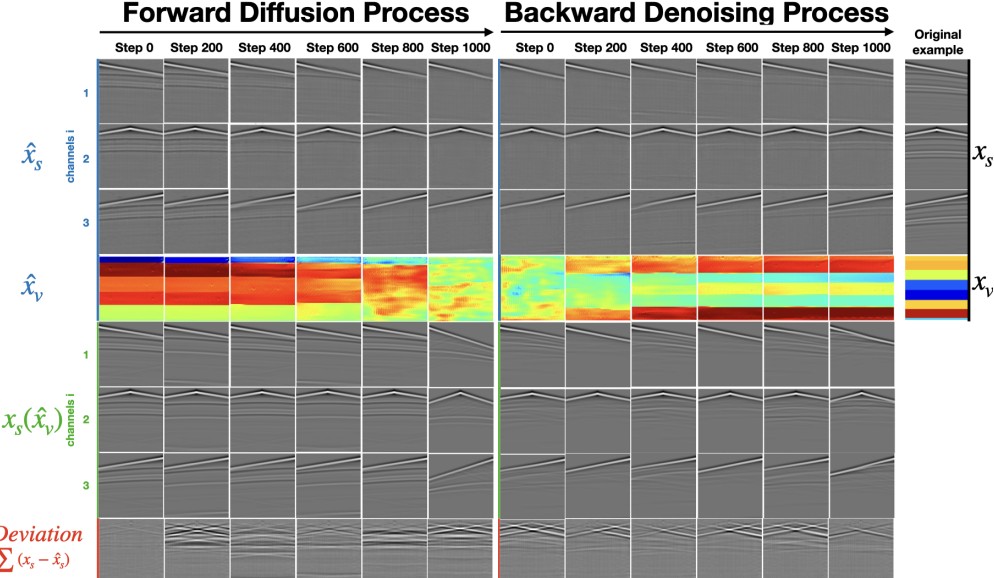

Figure 11: **Visualization of deviation from PDE during diffusion on FVB:** Seismic data comparison of the FVB example at different timesteps during the forward diffusion (left half) and reverse denoising processes (right half). The top three rows show seismic channels $\hat{x}_s$ generated by the joint diffusion model, while the fourth row shows the generated velocity map $\hat{x}_v$. Rows 5-7 show the synthetic seismic data $x_s(\hat{x}_v)$, simulated using a finite difference PDE solver on the generated $\hat{x}_v$. The last row shows the deviation from the governing PDE, visualized as the channel-stacked difference between $x_s(\hat{x}_v)$ and $\hat{x}_s$. Noise increases the deviation from the PDE during the forward diffusion, and the reverse process reduces discrepancies.

## A.3 COMBINED DATASET RESULTS

To assess the generalization capability of WaveDiffusion model across various geological structures, we trained the autoencoder and joint diffusion on a combined dataset that includes all the subsets.

The results on this combined dataset were consistent with those observed in the individual subsets, shown in Figure 13. The joint diffusion model effectively generated seismic data and velocity maps that adhered to the wave equation, regardless of the underlying geological configuration. The diffusion process successfully captured the shared latent space across different geological settings, though the features of different subsets are fused to some extent due to the shared latent space.

## A.4 JOINT GENERATION EXAMPLES

We present additional examples generated using the trained WAVEDIFFUSION model, emphasizing its ability to preserve symmetry and structural consistency across different datasets, particularly the FlatVel_B (FVB) subset.

As shown in the top two rows of Figure 14 (FVB subset), the seismic data inputs exhibit perfect symmetry with respect to a central vertical plane. Correspondingly, the generated velocity maps maintain this symmetry, demonstrating the model's ability to respect the geometric properties inherent in the input data. These results underscore the robustness of the WAVEDIFFUSION model in preserving geometric consistency and generating symmetric solutions, which is crucial for inversion tasks that rely on such properties.

Beyond the FVB subset, the middle and lower rows in Figure 14 present results from additional datasets, including the FFB and CVB subsets. These examples showcase the diversity and fidelity

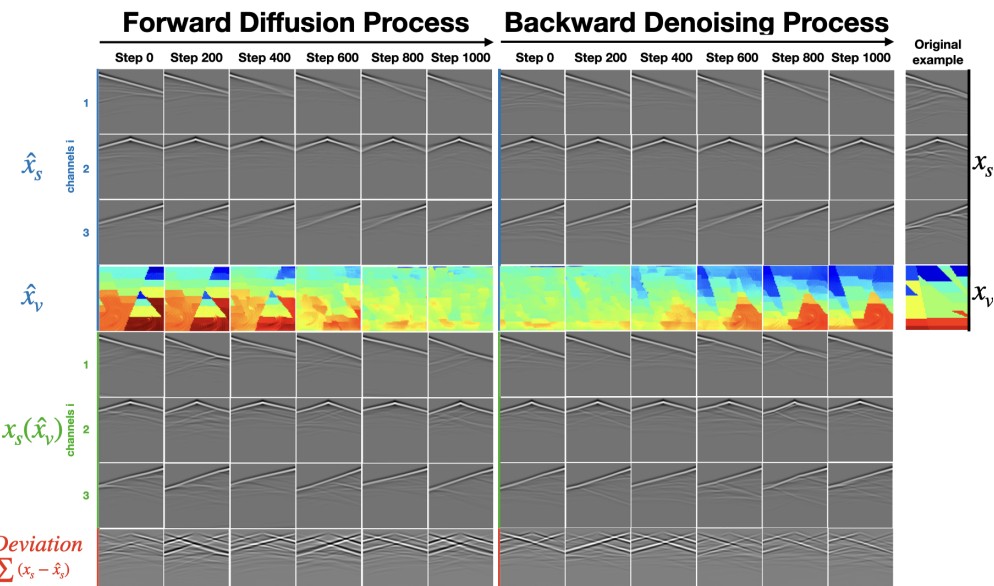

Figure 12: **Visualization of deviation from PDE during diffusion on FFB:** Seismic data comparison of the FFB example at different timesteps during the forward diffusion (left half) and reverse denoising processes (right half). The top three rows show seismic channels $\hat{x}_s$ generated by the joint diffusion model, while the fourth row shows the generated velocity map $\hat{x}_v$. Rows 5-7 show the synthetic seismic data $x_s(\hat{x}_v)$, simulated using a finite difference PDE solver on the generated $\hat{x}_v$. The last row shows the deviation from the governing PDE, visualized as the channel-stacked difference between $x_s(\hat{x}_v)$ and $\hat{x}_s$. Noise increases the deviation from the PDE during the forward diffusion, and the reverse process reduces discrepancies.

of the generated seismic-velocity pairs. The velocity maps capture layered structures and variations consistent with seismic data inputs, highlighting the model's capacity to handle a wide range of input patterns while maintaining physical and structural coherence.

Overall, the examples in Figure 14 demonstrate the effectiveness of our approach across different data distributions, confirming that the joint diffusion model can generalize well and produce reliable results for seismic data-velocity map generation tasks.

## A.5 NOISY INPUT OF BOTH MODALITIES

In this experiment, we tested the joint diffusion model's capability to handle noisy inputs and compared it with the reconstruction ability of the autoencoder alone. We started with clean seismic data-velocity pairs, representing the ideal (target) outputs for this experiment, as shown in the first column of Figure 16. Gaussian noise was then applied to both the seismic and velocity images, creating noisy versions of the input data (second column), simulating real-world scenarios where data may be degraded.

These noisy images were then fed into the encoders, producing a latent vector $z$ that was deviated from its true position in latent space due to the added noise. This noisy $z$ vector was first processed through the VQ layer and decoded directly, without applying the diffusion process. The results, shown in the third column, reveal that the autoencoder-only reconstruction suffers from noticeable distortions in both the seismic events and velocity layers. Artifacts, especially in the velocity reconstructions, indicate that the autoencoder alone is insufficient for restoring noisy inputs accurately.

To further improve the reconstruction, we passed the noisy latent $z$ vector into the joint diffusion model. In this step, the joint diffusion process progressively denoised $z$ through a series of steps, gradually refining it back toward the true latent representation. After denoising, the refined $z$ vector was decoded through the VQ and decoder, yielding final reconstructed images shown in the last column. These results demonstrate that the joint diffusion model effectively removes noise, restoring both the seismic events and the velocity layers to their correct shapes without artifacts.

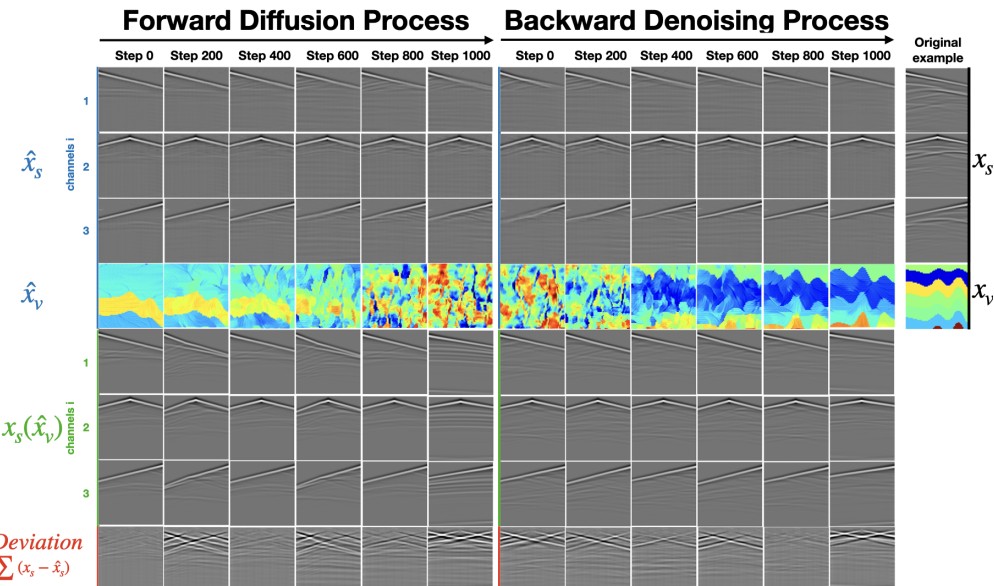

Figure 13: **Visualization of deviation from PDE during diffusion on combined datasets:** Seismic data comparison of the combined OpenFWI datasets example at different timesteps during the forward diffusion (left half) and reverse denoising processes (right half). The top three rows show seismic channels $\hat{x}_s$ generated by the joint diffusion model, while the fourth row shows the generated velocity map $\hat{x}_v$. Rows 5-7 show the synthetic seismic data $x_s(\hat{x}_v)$, simulated using a finite difference PDE solver on the generated $\hat{x}_v$. The last row shows the deviation from the governing PDE, visualized as the channel-stacked difference between $x_s(\hat{x}_v)$ and $\hat{x}_s$. Noise increases the deviation from the PDE during the forward diffusion, and the reverse process reduces discrepancies.

This experiment highlights a key discovery: the joint diffusion model is not only capable of inversion but also robust in denoising. By refining the noisy latent representation, the diffusion process ensures that the reconstructed outputs match the target seismic-velocity pairs, even when the input data contains significant noise. This ability to correct and clean degraded data underscores the model's value in practical applications involving noisy seismic measurements.

A.6    INVERSION COMPARISON TO VELOCITYGAN AND UPFWI

Table 6: **Comparison of VelocityGAN and UPFWI.** Metrics include RMSE, MAE, and SSIM for different setups and datasets.

| Dataset | Setup | VelocityGAN | | | UPFWI | | |
|---|---|---|---|---|---|---|---|
| | | **RMSE** | **MAE** | **SSIM** | **RMSE** | **MAE** | **SSIM** |
| CVB | PureGen | 0.4556 | 0.2971 | 0.5186 | 0.4616 | 0.3182 | 0.5007 |
| | Gen+1% | 0.3098 | 0.1778 | 0.6240 | 0.3935 | 0.2424 | 0.6177 |
| | 1%Only | 0.4842 | 0.3650 | 0.3583 | 0.4887 | 0.3612 | 0.3945 |
| | OpenFWI | 0.2611 | 0.1268 | 0.6962 | 0.3179 | 0.1777 | 0.6614 |
| FFB | PureGen | 0.2545 | 0.1678 | 0.6799 | 0.2847 | 0.1873 | 0.6541 |
| | Gen+1% | 0.1635 | 0.0917 | 0.6878 | 0.1880 | 0.1011 | 0.6816 |
| | 1%Only | 0.3130 | 0.2416 | 0.5297 | 0.3150 | 0.2315 | 0.5360 |
| | OpenFWI | 0.1553 | 0.0925 | 0.7552 | 0.2220 | 0.1416 | 0.6937 |

Table 6 summarizes the performance of VelocityGAN and UPFWI trained on WAVEDIFFUSION-generated samples under different setups, evaluated on the CVB and FFB datasets.

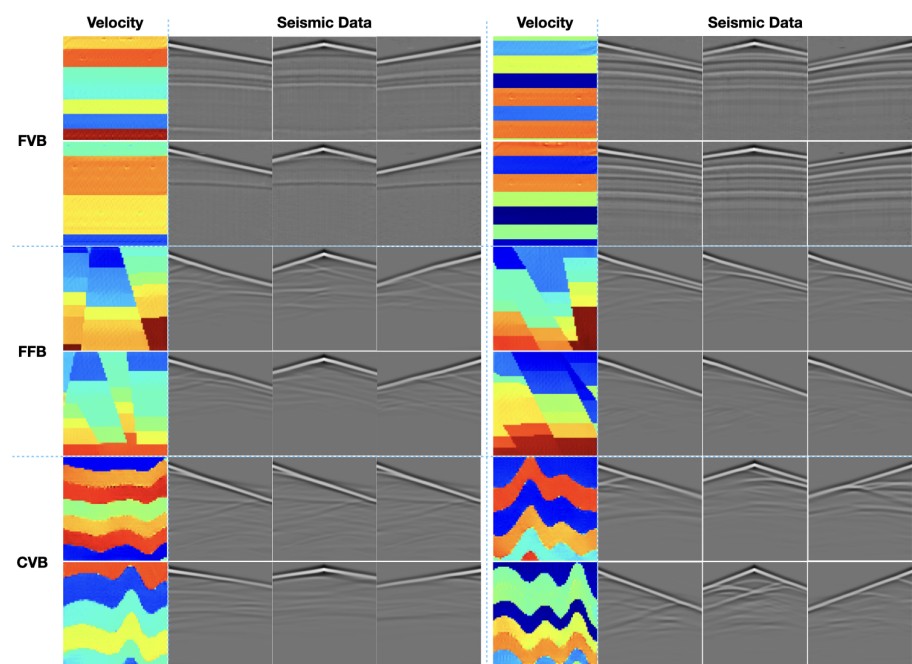

Figure 14: **Generated Examples from Various Subsets.** Examples from the FVB, FFB, and CVB subsets. The first two rows show symmetric seismic data inputs and corresponding velocity maps (FVB subset), illustrating the model's ability to preserve symmetry. The middle and bottom rows display more diverse examples from FFB and CVB subsets, highlighting the model's generalization capability.

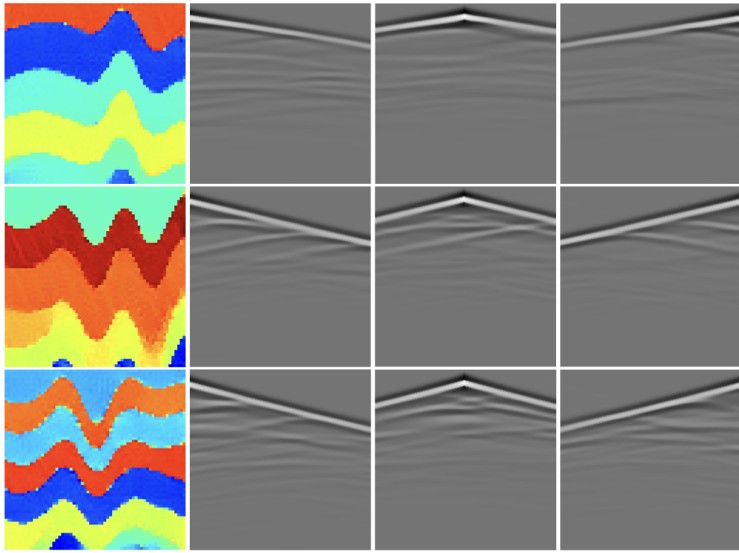

Figure 15: **Generated Examples from CVB Subset Using the One-in-two-out Joint Diffusion.** Three examples from the CVB subset. Each row shows one velocity (column 1) and the corresponding seismic data (columns 2-4) generations.

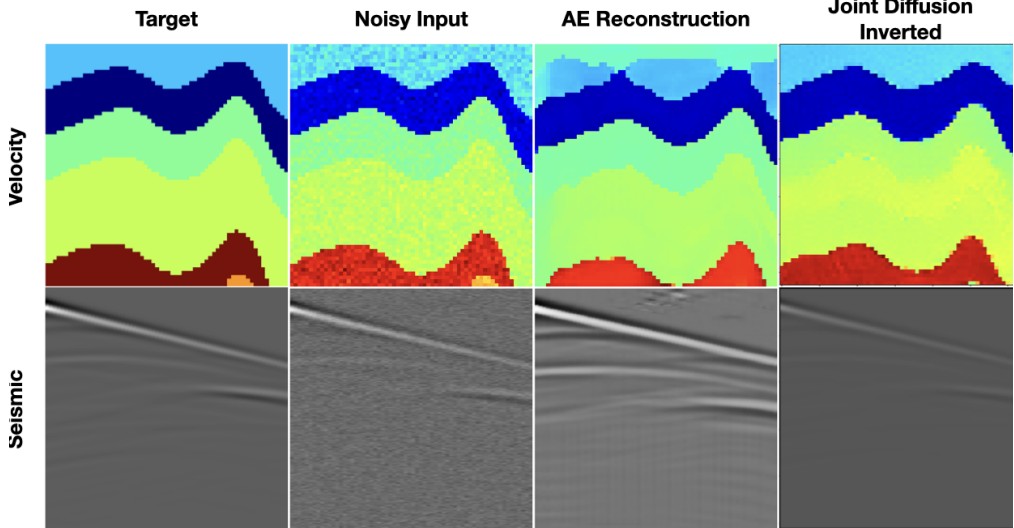

Figure 16: **Reconstructions with Noisy Inputs.** From left to right: target clean images, Gaussian noisy inputs, autoencoder reconstructed images from noisy inputs, and joint diffusion reconstructions. When noisy data is input, the autoencoder only reconstructs deformed images with artifacts, while the diffusion process refines the noisy inputs back to their target images.

Table 7: **Comparison of InversionNet performance on CVB and FVB datasets.** Performance metrics for different training setups.

| Dataset | Setup | RMSE | MAE | SSIM |
|---|---|---|---|---|
| CVB | PureGen | 0.4798 | 0.3078 | 0.5204 |
| | Gen+1% | 0.3258 | 0.1976 | 0.6293 |
| | 1%Only | 0.4915 | 0.3833 | 0.3625 |
| | OpenFWI | 0.2801 | 0.1624 | 0.6661 |
| FVB | PureGen | 0.2766 | 0.1723 | 0.6895 |
| | Gen+1% | 0.1737 | 0.0893 | 0.8532 |
| | 1%Only | 0.4045 | 0.2884 | 0.4967 |
| | OpenFWI | 0.0909 | 0.0417 | 0.9402 |

**Gen+1%:** Adding 1% real data to the generated samples significantly boosts performance for both solvers, with VelocityGAN achieving an SSIM of 0.6240 and UPFWI reaching 0.6177 on the CVB dataset. This highlights the value of combining generated and real data.

**PureGen and 1 %Only**: Models trained solely on generated samples (**PureGen**) or 1% real data (**1%Only**) show limited performance, with SSIM scores below 0.53. This demonstrates the importance of real data for improving generalization.

**Comparison on OpenFWI**: When trained on the full OPENFWI dataset, both VelocityGAN and UPFWI achieve strong results, with VelocityGAN slightly outperforming UPFWI in SSIM across both datasets.

A.7 ADDITIONAL INVERSIONNET COMPARISONS

The full comparison of InversionNet performance across CVB and FVB datasets under the same experimental setups is shown in Table 7. Below are the key observations:

**CVB**: InversionNet trained on PureGen samples performs moderately well but trails behind the OPENFWI baseline. Combining generated samples with 1% of real data (Gen+1%) results in notable improvement, outperforming the 1%Only setup by a significant margin.

**FVB**: On the FVB dataset, PureGen samples demonstrate competitive results, and the Gen+1% setup achieves near-baseline performance. The 1%Only setup, in contrast, shows significantly worse results, highlighting the limitations of minimal data availability.

The visual comparisons for the CVB and FVB datasets are shown in Figure 17.

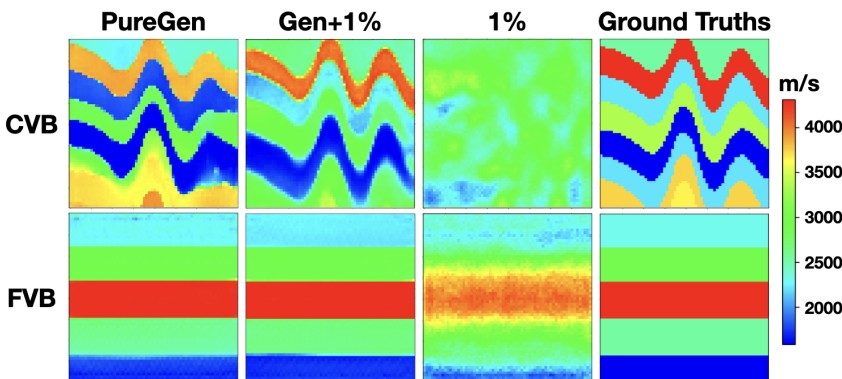

Figure 17: **InversionNet performance visualization.** Predictions using (column 1) WaveDiffusion generated samples, (column 2) WaveDiffusion generated + 1% OpenFWI samples, and (column 3) 1% OpenFWI samples compared to ground truth (column 4) across CVB, FFB, and FVB datasets.

## A.8 PERFORMANCE COMPARISON OF FWI WITH NOISY SEISMIC DATA

This experiment evaluates the robustness of WAVEDIFFUSION in inverting noisy seismic data compared to InversionNet and VelocityGAN. By introducing Gaussian noise (mean=0, std=0.05) to clean seismic data, we simulate a realistic scenario that reflects challenges in field data acquisition.

Our WAVEDIFFUSION model was tasked with recovering both velocity maps and noise-free seismic data from the noisy seismic inputs. In contrast, InversionNet and VelocityGAN were tested only on recovering velocity maps. Importantly, none of the models were trained on noisy data, making this a stringent test of robustness.

The results in Table 8 and Figure 18 highlight the significant advantage of WAVEDIFFUSION over the baselines. WAVEDIFFUSION achieves the lowest MAE and RMSE and the highest SSIM, demonstrating its ability to effectively handle noisy data.

Table 8: **Performance Metrics for Noisy Seismic Data Inversion.** Evaluation based on Gaussian noisy seismic data as input and velocity maps as output on the CVB dataset.

| Model | MAE | RMSE | SSIM |
|---|---|---|---|
| InversionNet | 0.9069 | 1.0701 | 0.2604 |
| VelocityGAN | 0.4913 | 0.6916 | 0.3231 |
| WaveDiffusion (Ours) | **0.2227** | **0.3776** | **0.6142** |

Despite being trained only on clean data, WAVEDIFFUSION demonstrates exceptional robustness to noisy inputs. Its joint diffusion process effectively denoises the seismic data during inversion, recovering high-fidelity velocity maps and seismic data that align with the physical constraints of the wave equation.

In contrast, both InversionNet and VelocityGAN fail to handle noisy seismic data, as evidenced by significantly higher errors (MAE and RMSE) and lower structural similarity (SSIM). This limitation stems from their image-to-image mapping architectures, which lack an inherent denoising mechanism.

These results underline the practical utility of WAVEDIFFUSION in real-world scenarios where data noise is inevitable. By refining noisy latent representations through diffusion, our model not only recovers accurate velocity maps but also reconstructs noise-free seismic data, setting it apart from traditional mapping-based methods.

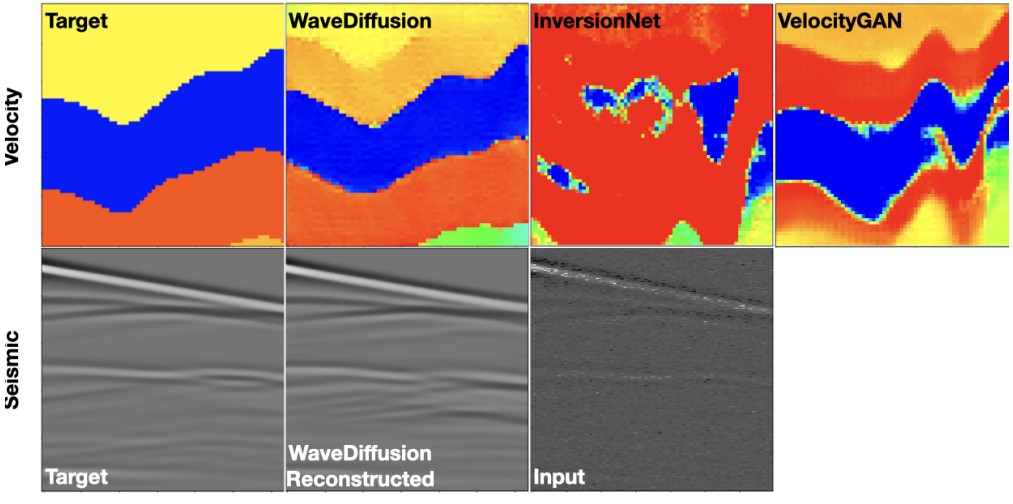

Figure 18: **FWI with Noisy Seismic Data Input.** From left to right: target clean images, WAVED-IFFUSION FWI results from noisy inputs, InversionNet results, and VelocityGAN results. Inversion-Net and VelocityGAN fail to handle noisy inputs, while WAVEDIFFUSION robustly refines the noisy seismic data to generate accurate velocity maps and noise-free seismic data.

