# OpenReview forum: "WaveDiffusion: Exploring Full Waveform Inversion via Joint Diffusion in the Latent Space"
_ICLR.cc/2025/Conference — Submitted to ICLR 2025_

### Official Review · Reviewer_rGTJ · 2024-10-23

**Soundness:** 3
**Presentation:** 2
**Contribution:** 2
**Rating:** 3
**Confidence:** 5

**Summary:**

This paper deals with the problem of full waveform inversion.
There are two mechanisms that the paper proposes.
1. The paper uses the same latent space for both the model and the data
2. They train a diffusion model in the latent space. Such a diffusion model can therefore generate a plethora of models and their data.

Results look reasonable even though the models that are being trained on are very simple.

**Strengths:**

The idea of using a joint feature space is good and then using a diffusion model on this space is also a good idea. The results are interesting and it seems that the approach works for the models in the data base.

**Weaknesses:**

Unfortunately, the idea of using an AE with common feature spaces for data and model is not new. See https://paperswithcode.com/paper/paired-autoencoders-for-inverse-problems
This is the main problem that the paper have. I understand that in this fast moving field some papers are missed but in this case, the work that was already done makes much of the paper not relevant.
I would recommend the authors to withdraw the paper, concentrate of the diffusion aspect of the paper and resubmit to a different venue.

**Questions:**

The interesting parts of the paper are actually hiding towards the end.

1. How do you actually do coarse to fine?

2. Given some data $d$ how to you use diffusion to find an appropriate model

I would recommend re-writing the paper with section 3.3 in mind. Since training a dual AE is not very innovative and using diffusion of the latent space is not very innovating, the innovation is exactly what you do in 3.3. You could easily develop it to a full paper.

---

> ### Author Response · Authors · 2024-11-19
>
> **Weakness: Unfortunately, the idea of using an AE with common feature spaces for data and model is not new. See https://paperswithcode.com/paper/paired-autoencoders-for-inverse-problems This is the main problem that the paper have. I understand that in this fast moving field some papers are missed but in this case, the work that was already done makes much of the paper not relevant. I would recommend the authors to withdraw the paper, concentrate of the diffusion aspect of the paper and resubmit to a different venue.**
>
> Thank you for bringing this reference to our attention. We will ensure it is cited appropriately in our revised manuscript. However, we respectfully disagree with the recommendation to withdraw the paper, as the **primary contribution of our work** is not the dual autoencoder architecture but rather the **joint diffusion process in a shared latent space**. The dual autoencoder serves only as a foundational step, enabling the creation of a shared latent representation for the two modalities, seismic data and velocity maps. The key novelty of our work lies in the use of this shared latent space for a joint diffusion process that refines the generated data pairs while maintaining physical consistency.
>
> To further illustrate this, we conducted experiments with a modified autoencoder architecture that uses a single encoder and two decoders (one-in-two-out configuration) in the first stage. This setup demonstrates the flexibility of our approach, where the joint diffusion process continues to perform effectively, preserving the discoveries outlined in the paper. The generation results for this modified architecture are available at https://bit.ly/4hVLg1q.
>
> Additionally, the one-in-two-out autoencoder configuration allows for direct application to standard FWI tasks, as shown in our response to Reviewer 1caw. In this setup, seismic data alone is used as input to the model, and the joint diffusion process generates both seismic and velocity outputs, making the approach suitable for practical applications. Example results for this experiment can be found at https://bit.ly/4eBIJX2.
>
> In summary, while we acknowledge the existence of prior work on paired autoencoders, our contribution is orthogonal to this line of research. We focus on the integration of joint diffusion into the latent space, which we believe is a novel and significant advancement in the field. We hope this clarification and the new experimental results address your concerns regarding the relevance and novelty of our work.
>
> **Question 1: How do you actually do coarse to fine?**
>
> Thank you for your question. We achieve coarse-to-fine refinement through the **latent space joint diffusion process**. Starting with approximate representations, the process iteratively refines these by scoring the deviation from the governing PDE in the latent space. The forward diffusion process perturbs the latent vector progressively, while the backward diffusion process denoises and aligns it to physically valid solutions. This approach ensures a gradual improvement, moving from coarse approximations to accurate representations governed by the PDE, adhering to the physical constraints of the problem.
>
> **Question 2: Given some data d how to you use diffusion to find an appropriate model?**
>
> As noted in our response to *Weakness 1 to Reviewer 1caw* and to your raised *Weakness* above, we have conducted a new experiment to address this question. The experiment demonstrates how our framework finds an appropriate velocity model given specific seismic data **d**.
>
> To address this task, we utilize a "one-in-two-out" configuration within the autoencoder part of our framework. In this setup: only seismic data (**d**) is input to the encoder, generating a latent representation solely contributed by seismic input. This latent vector is then refined through the joint diffusion process, iteratively denoising and aligning it with the physical constraints.
> Finally, the refined latent vector is passed through individual VQ layers and decoders, generating both seismic and velocity outputs.
> This setup enables the inversion of seismic data to produce velocity maps, making our model directly applicable to standard FWI tasks. Example results from this experiment are available at http://bit.ly/4eBIJX2. We will incorporate these results into the revised manuscript to demonstrate this capability in greater detail.

---

> > ### Author Response · Authors · 2024-11-22
> > **Reminder: Follow-up on Rebuttal Responses**
> >
> > Dear Reviewer,
> >
> > We hope this message finds you well. We wanted to kindly remind you that the author-reviewer discussion period is ongoing, and we have posted our responses and revisions over two days ago. Your thoughtful feedback has been immensely valuable in shaping our work, and we greatly appreciate your time and effort in reviewing our manuscript.
> >
> > We have carefully addressed your concerns and answered your questions. If you have any further questions or would like clarification on our responses, we are here and ready to address them promptly. Your input is crucial for refining and improving our manuscript, and we deeply value your insights.
> >
> > We understand your time is valuable, and we sincerely thank you for your dedication to this process. Should you have any additional thoughts or concerns, please do not hesitate to reach out.
> >
> > Thank you again for your attention and support.
> >
> > Best regards,
> > -Authors

---

> ### Comment · Reviewer_rGTJ · 2024-11-22
>
> While the authors have attempted to answer my concerns, the paper needs many changes to reflect
> 1. The novelty in the diffusion process (and not in the auto-encoder)
> 2. The inversion process (getting a model given d)
>
> My rating stands, the authors need to rewrite the paper focusing on the diffusion process and the inversion strategy. The paper as is, lacks the needed novelty. I am sure that they can make it better for future conference

---

> ### Author Response · Authors · 2024-11-23
> **Paper revision to address your concerns on novelty and inversion process**
>
> Dear Reviewer,
>
> Thank you for your detailed and thoughtful feedback on our manuscript. We sincerely value this opportunity to contribute to the conference and are committed to improving our work to meet the high standards of innovation and clarity expected. Thus, we have uploaded a **revised version** of our manuscript. Your critique has been invaluable in guiding our revisions, and we are confident that the updates made reflect our commitment to addressing your concerns.
>
> Below are the key updates specific to your suggestions:
>
> **Section 4.6 on page 9 (WaveDiffusion for conventional FWI):**
>
> We have added a new experiment demonstrating how our WaveDiffusion model can perform conventional FWI when only seismic data is provided. This addresses the concern about the inversion process (i.e., obtaining a velocity model given seismic data). The experiment uses a “one-in-two-out” configuration of the autoencoder, where only seismic data is input, and the diffusion process generates both seismic and velocity outputs. Results show that our joint diffusion approach effectively reconstructs velocity maps from seismic inputs, providing a clear demonstration of our inversion strategy.
>
> **Section 3.2 on page 4 (Clarification of the Role of Dual Autoencoder):**
>
> We have explicitly cited the suggested references and clarified that the dual autoencoder is not the primary contribution of our paper. Instead, it is a preliminary step to establish the shared latent space required for the diffusion process. The true novelty lies in the joint diffusion process, which refines latent representations to adhere to the governing PDE. This distinction is clearly stated in the revised manuscript, with Section 3.3 and the subsequent experiments highlighting the core contributions.
>
> **Highlighting the Novelty of Diffusion and Inversion:**
>
> Throughout the paper, we have emphasized that our main contribution lies in the diffusion process and its application to solving FWI problems in a novel generative framework. By restructuring and refocusing key sections, we have aligned the narrative to better reflect the innovative aspects of the joint diffusion process and its inversion capabilities.
>
> We genuinely believe that these changes align our work more closely with your suggestions and enhance its contribution to the field. We are excited about the opportunity to present our research at this conference, as we feel it introduces a unique and valuable perspective on FWI.
>
> If there are still areas that you find unsatisfactory or needing improvement, we would deeply appreciate further guidance. Your feedback is critical to helping us refine our work and make it as impactful as possible.
>
> Thank you again for your time and effort in reviewing our submission. We look forward to your insights.
>
> Best regards,
>
> The Authors

---

> > ### Author Response · Authors · 2024-11-23
> > **Extra experiment to address your concerns of novelty and inversion process**
> >
> > Dear Reviewer,
> >
> > We hope this message finds you well. As part of our continued efforts to address all reviewer feedback comprehensively, we recently added a new experiment to the revised manuscript, showcasing the robustness of *WaveDiffusion* in handling noisy input seismic data.
> >
> > This new experiment (Appendix A.8) highlights the capability of *WaveDiffusion* to invert seismic data corrupted with Gaussian noise, recovering both high-quality velocity maps and noise-free seismic data. Notably, this was achieved without additional training on noisy datasets. In contrast, baseline methods like VelocityGAN and InversionNet struggled significantly under the same conditions.
> >
> > The results are detailed in the revised manuscript, and visual and quantitative comparisons are available via the following links:
> >
> > The visual comparisons are shown in the figure at https://bit.ly/3V5k0Uc, and the quantitative results are detailed in the table at https://bit.ly/4f0rAqB. The new experiment and results are added to the latest revised manuscript on Page 19 A. 8 section.
> >
> > We believe this experiment further underscores the versatility and real-world applicability of *WaveDiffusion*, addressing scenarios where data noise is a common challenge in FWI tasks.
> >
> > If you have any additional suggestions or questions about this experiment, we would be delighted to address them. Thank you for your time and valuable insights, which continue to guide the improvements to our work. If there are still areas that you find unsatisfactory or needing improvement, we would deeply appreciate further guidance. Your feedback is critical to helping us refine our work and make it as impactful as possible.
> >
> > Best regards,
> >
> > The authors

---

> > > ### Author Response · Authors · 2024-11-25
> > > **Response to Your Feedback and Request for Further Guidance**
> > >
> > > Dear Reviewer rGTJ,
> > >
> > > Thank you for taking the time to review our manuscript and provide your detailed feedback. We deeply appreciate your insights, which have been invaluable in shaping the revised version of our paper (uploaded on Nov. 23). We have carefully addressed your concerns and made significant revisions to the manuscript to clarify the novelty of our work and strengthen its focus on the diffusion process and inversion strategy. Below, we outline your original comments, our responses, and the corresponding revisions. We kindly ask for your feedback on whether these revisions address your concerns, and if not, how we can further improve the paper.
> > >
> > > **Your Comment: The novelty in the diffusion process (and not in the autoencoder) should be highlighted.**
> > >
> > > **Our Response:** We agree with your suggestion that the diffusion process is the key novelty of our work. In the revised manuscript, we have shifted the emphasis to highlight the joint diffusion process as the central contribution. The dual autoencoder is explicitly described as a supporting step to establish a shared latent space, rather than the primary focus.
> > >
> > > **Our Revision to the Manuscript:** ***Similar to the response to the reviewer A1pp***, the novelty and importance of the joint diffusion process are emphasized throughout the paper, particularly in:
> > >
> > > **Section 3.3:** The **joint diffusion process** is presented as the core innovation, enabling the model to refine latent representations in a shared space to adhere to physical constraints governed by PDEs.
> > >
> > > **Section 4.3–4.6:** Experimental results demonstrate the efficacy and uniqueness of the **joint diffusion framework**. For instance, Section 4.6 applies the **joint diffusion model** to solve a conventional FWI problem where only seismic data is provided.
> > >
> > > **Appendices A.2–A.8:** Additional experiments highlight the robustness, physical consistency, and versatility of the **joint diffusion process** across various scenarios.
> > >
> > > These revisions ensure that the **joint diffusion process** is firmly presented as the primary contribution of the paper, minimizing the role of the autoencoder, which serves only as a means to facilitate the joint diffusion.
> > >
> > > **Your Comment: The inversion process (getting a model given d) needs more focus.**
> > >
> > > **Our Response:** We recognize the importance of addressing this aspect and have conducted additional experiments to showcase the application of our **joint diffusion** framework for conventional FWI tasks. Specifically, in **Section 4.6 and Appendix A.8**, we introduce the “one-in-two-out” architecture, where only seismic data (d) is input to the encoder. This experiment demonstrates how the model can invert seismic data to generate accurate velocity maps, satisfying the governing PDE.
> > >
> > > **Our Revision to the Manuscript:**
> > >
> > > **Section 4.6:** Details the “one-in-two-out” experiment, which applies the **joint diffusion model** to invert seismic data into velocity maps. This setup reflects real-world FWI scenarios and highlights the practicality of our approach.
> > >
> > > **Appendix A.8:** Includes additional experiments with noisy seismic data as input, showcasing the robustness of our **joint diffusion framework** in challenging scenarios.
> > >
> > > These experiments explicitly demonstrate how our **joint diffusion model** handles inversion tasks, providing clear evidence of its capabilities and novelty.
> > >
> > > **Request for Feedback**
> > >
> > > Given the extensive revisions and new experiments added to address your concerns, we kindly ask if the current manuscript adequately reflects the novelty of the diffusion process and the inversion strategy. If there are any remaining gaps or areas for improvement, we would like to hear your suggestion on the reason. We would greatly appreciate your further feedback. Your guidance is invaluable to us as we strive to enhance the quality and impact of our work.
> > >
> > > Thank you again for your time and thoughtful review.
> > >
> > > Best regards,
> > >
> > > The Authors

---

> > > > ### Author Response · Authors · 2024-11-28
> > > > **Second Reminder to Reviewer rGTJ**
> > > >
> > > > Dear Reviewer rGTJ,
> > > >
> > > >
> > > > We hope this message finds you well. We are writing to kindly follow up on the latest revised version of our manuscript, which was uploaded on November 23. We have made extensive revisions to address your earlier comments, particularly emphasizing the novelty and contributions of the joint diffusion process.
> > > >
> > > >
> > > > We understand and respect your decision to maintain your score, but your insights and suggestions on the current version of the manuscript are invaluable to us. We are eager to hear any further comments you might have regarding areas that could be improved or clarified. Your feedback is crucial for us to refine the paper and ensure its quality and impact.
> > > >
> > > >
> > > > Thank you again for your time and effort in reviewing our work. We deeply appreciate your engagement and look forward to any additional suggestions you may have.
> > > >
> > > >
> > > > Best regards,
> > > >
> > > > The authors

---

> > > > > ### Author Response · Authors · 2024-12-01
> > > > > **Third Reminder to Reviewer rGTJ: Follow-up on the Revised Manuscript**
> > > > >
> > > > > Dear Reviewer rGTJ,
> > > > >
> > > > > We hope this message finds you well.
> > > > >
> > > > > We are kindly following up to ask if you have any additional comments on the latest revised version of our manuscript, uploaded on November 23. Your feedback is very important to us as we strive to improve the quality and clarity of our work.
> > > > >
> > > > > Thank you again for your time and effort, and we look forward to any further suggestions you may have.
> > > > >
> > > > > Best regards,
> > > > >
> > > > > The authors

---

> > > > > > ### Author Response · Authors · 2024-12-02
> > > > > > **Fourth Reminder to Reviewer rGTJ: Follow-up on the Revised Manuscript**
> > > > > >
> > > > > > Dear Reviewer rGTJ,
> > > > > >
> > > > > > We hope this message finds you well.
> > > > > >
> > > > > > We are kindly following up to ask if you have any additional comments on the latest revised version of our manuscript, uploaded on November 23. Your feedback is very important to us as we strive to improve the quality and clarity of our work.
> > > > > >
> > > > > > Thank you again for your time and effort, and we look forward to any further suggestions you may have.
> > > > > >
> > > > > > Best regards,
> > > > > >
> > > > > > The authors

---

> > > > > > > ### Author Response · Authors · 2024-12-02
> > > > > > > **Last Day Reminder to Reviewer rGTJ: Follow-up on the Revised Manuscript**
> > > > > > >
> > > > > > > Dear Reviewer rGTJ,
> > > > > > >
> > > > > > > We hope this message finds you well.
> > > > > > >
> > > > > > > We are reaching out to kindly remind you that today is the **last day** of the discussion session. We would like to kindly ask you for your follow-up on the latest revised version of our manuscript, uploaded on November 23. We would greatly appreciate any additional comments or suggestions you may have on the current version.
> > > > > > >
> > > > > > > Thank you for your time and thoughtful feedback. We look forward to hearing from you.
> > > > > > >
> > > > > > > Best regards,
> > > > > > >
> > > > > > > The authors

---

### Official Review · Reviewer_STHJ · 2024-11-01

**Soundness:** 3
**Presentation:** 4
**Contribution:** 3
**Rating:** 6
**Confidence:** 3

**Summary:**

Full waveform inversion (FWI) is a seismic imaging technique that traditionally reconstructs the subsurface velocity model by iteratively comparing observed and predicted seismic data. More recently, machine learning-based approaches would solve FWI by treating it as an image-to-image translation problem. Furthermore, generative diffusion models mainly treated FWI as a conditional generation problem where the velocity map is generated from a given seismic data. This paper offers a new perspective on FWI by considering it as a joint generative process. Namely, the paper considers whether the two modalities -- seismic data and velocity map -- can be generated simultaneously. Two key steps are proposed: first, a dual autoencoder encodes the two modalities in a shared latent space that provides a coarse approximation of the wave equation solution. Second, a diffusion process in the latent space refines the coarse latent representations which are later decoded into seismic data and velocity maps. In contrast to seismic-velocity pairs generated by the conditional models which often lack physical consistency, the jointly generated pairs approximately satisfy the governing PDE without any additional constraint. The paper's main goal is to offer a new perspective by extending FWI from a conditional generation problem to a joint generation problem.

**Strengths:**

- The proposed paper is well-organized and the idea is clearly presented.
- The paper offers a new perspective on the FWI generation problem by simultaneously generating two modalities -- seismic data and velocity maps -- from the shared latent space. This is a novel idea in contrast to the existing related work which treats these two modalities separately.
- Treating seismic data and velocity maps separately limits the ability to generate physically consistent seismic-velocity pairs. In contrast, jointly generating these modalities makes them approximately consistent with the governing PDE that describes the relationship between them.
- The extensive experiments confirm the soundness of the proposed method and show that the jointly generated seismic-velocity pairs can be a useful supplement to real training data.

**Weaknesses:**

I think there are three main problems in the experiments:
- The method wasn't compared to any existing conditional generative methods. There is even a section 4.2.4. that compares separate vs. joint diffusion but there the separate diffusion was the same model as for the joint diffusion but with a single branch kept active and the latent space no longer shared. I think it would be useful to see how the proposed method compares to the existing methods (e.g., [1]) both in terms of the diversity of the generated data and the performance of the reconstruction methods when trained on the generated data.
- The results might also differ based on a different reconstruction method other than InversionNet (e.g., [2] and/or [3]). I think it would be beneficial to add at least one additional data-driven solver.
- Some of the experiments in the results section do not seem to be realistic. (see more in the questions section)

---

[1] F. Wang, X. Huang, and T. A. Alkhalifah. "A prior regularized full waveform inversion using generative diffusion models." IEEE Transactions on Geoscience and Remote Sensing, 61:1-11, 2023.

[2] P. Jin, X. Zhang, Y. Chen, S. Huang, Z. Liu, and Y. Lin. "Unsupervised learning of full-waveform inversion: Connecting CNN and partial differential equation in a loop." ICLR, 2022.

[3] Z. Zhang, Y. Wu, Z. Zhou, and Y. Lin. "VelocityGAN: Subsurface velocity image estimation using conditional adversarial networks." In 2019 IEEE Winter Conference on Applications of Computer Vision (WACV), 2019, pp. 705-714.

**Questions:**

1. Could you comment on the comparison with the existing conditional generative models? Why didn't you compare to any of the existing methods at least in 4.2.4 section?
2. Could you comment on the choice of the reconstruction method? I think it would be beneficial to add at least one additional data-driven solver. It would be interesting to see how the reconstruction methods work with data generated by different generative models.
3. The generative model was trained on the same OpenFWI dataset on which InversionNet was later evaluated. What is the amount of data your generative model should be trained with and how does it compare to the size of a dataset reconstruction methods (e.g., InversionNet) should be trained with? If the size of a dataset for reconstruction methods is satisfying what is the rationale of doing this? I think you should address the limitations of such a setup.
4. In continuation to the previous question, how realistic is the Gen+1\% case? In this case, you trained your generative model on the same data distribution as in the 1\% of the original dataset. If a real dataset is small, wouldn't it be more realistic to train your generative model with real data that differ from the distribution in the small dataset? Maybe a more realistic case would be to train the generative model on the two subsets and add 1\% of the third subset of OpenFWI. Could you comment on this? What are the implications of the existing setup for real-world applications of the method?

---

> ### Author Response · Authors · 2024-11-19
>
> **Reviewer STHJ**
>
> **Weakness 1: Lack of comparison to conditional generative methods.**
>
> Thank you for your suggestion. Our method is a joint diffusion generation framework without conditioning, distinct from conditional generative approaches like [1], which integrate finite difference modeling. Our focus is to find solutions in a shared latent space by scoring deviations from the wave equation, avoiding iterative forward modeling steps. Thus, a direct comparison to [1] is not feasible as the objectives and methodologies differ fundamentally. Our work aims to provide a novel perspective on FWI, emphasizing joint diffusion for physical consistency, rather than competing as a conditional generative model. We will clarify these distinctions in the revised manuscript.
>
> **Weakness 2: Need for additional reconstruction methods.**
>
> Thank you for this valuable suggestion. We agree that evaluating with alternative reconstruction methods strengthens our analysis. We expanded the experiments in Section 4.2.3 to include UPFWI and VelocityGAN alongside InversionNet. The results, detailed at https://bit.ly/3YYfTua, show slight performance improvements (SSIM +~2%) with these solvers, confirming our primary conclusion that generated samples enhance end-to-end mapping networks. We will integrate these results into the revised manuscript.
>
> **Weakness 3: Realism of experiments.**
>
> Thank you for your observation. InversionNet is used to evaluate our generated data's quality rather than outperform existing FWI baselines. For practical applications, we propose denoising as a key use case for our approach, particularly when dealing with noisy or incomplete data. By leveraging the joint diffusion process, our model can refine and reconstruct physically consistent seismic-velocity pairs, even under challenging data conditions. This will be emphasized in the revised manuscript.
>
> **Question 1: Comparison with conditional generative models.**
>
> Thank you for this insightful question. As noted in *Weakness 1*, our joint diffusion framework differs fundamentally from conditional generative models like [1], focusing on discovering solution spaces in a shared latent space. Conditional approaches rely on specific conditions and finite difference modeling. Given these fundamental differences, a direct comparison would not be equitable. Moreover, our primary goal is to introduce a novel perspective for FWI, focusing on joint diffusion's potential for refining data pairs in a physically consistent manner, rather than proposing a SOTA FWI solution. We will clarify these distinctions further in the revised manuscript to address this concern comprehensively.
>
> **Question 2: Alternative reconstruction methods.**
>
> As addressed in *Weakness 2*, we evaluated additional algorithms (UPFWI and VelocityGAN), confirming the robustness of our generated samples. The results are detailed at https://bit.ly/3YYfTua and will be added to the manuscript.
>
> **Question 3: Dataset overlap in training and evaluation.**
>
> Thank you for this question. We added an experiment demonstrating the **denoising capabilities** of our joint diffusion model. By adding Gaussian noise to both seismic data and velocity maps, we observed that direct autoencoder reconstruction from noisy latent vectors produced suboptimal outputs, with visible noise and distortions. However, applying the joint diffusion process refined the latent vectors, yielding clean and consistent reconstructions, as shown at https://bit.ly/4eDJAH0. This highlights our model's practical application for noisy or incomplete data, addressing real-world challenges. As mentioned in *Weakness 3*, InversionNet primarily serves as a quality evaluation tool, not a direct benchmark, in our study.
>
> We will clarify these points and include the denoising experiment in the revised manuscript.
>
> **Question 4: Realism of the Gen+1% setup.**
>
> Thank you for your insightful question. The Gen+1% setup evaluates the model’s ability to enhance a small dataset using supplementary in-distribution data, providing a controlled scenario for assessing performance. We agree that training on two subsets and fine-tuning with 1% of a third introduces a distribution shift, presenting an OOD challenge.
>
> While our study primarily focuses on in-distribution scenarios, reflecting many real-world applications, addressing OOD challenges is an exciting direction for future work. Techniques such as fine-tuning, domain adaptation, or transfer learning could be explored to extend our framework for handling unseen data distributions. Our joint diffusion model inherently refines latent representations toward physically valid solutions, providing a strong foundation for tackling OOD scenarios. These limitations and future directions will be highlighted in the revised manuscript.

---

> > ### Comment · Reviewer_STHJ · 2024-11-21
> > **Thank you for your response**
> >
> > Dear authors,
> >
> > thank you very much for your responses. Having carefully reviewed both the other reviewers' comments and your responses, I wish to maintain my current score.

---

> > > ### Author Response · Authors · 2024-11-22
> > > **Grateful for Your Feedback**
> > >
> > > Dear Reviewer,
> > >
> > > We sincerely thank you for your kind response and for the effort you’ve invested in reviewing our work. Your insights and constructive feedback have been incredibly valuable to us.
> > >
> > > If you have any additional suggestions or recommendations for enhancing our manuscript, we would be delighted to hear them.
> > >
> > > Once again, thank you for your thoughtful contributions and support.
> > >
> > > Best regards,
> > >
> > > Authors

---

### Official Review · Reviewer_A1pp · 2024-11-02

**Soundness:** 3
**Presentation:** 3
**Contribution:** 1
**Rating:** 3
**Confidence:** 5

**Summary:**

The paper introduces a new framework for Full Waveform Inversion (FWI) that uses a joint diffusion process in a shared latent space. This approach merges the bottlenecks of two separate autoencoders (one for seismic data and one for velocity maps) into a unified latent space.

**Strengths:**

The paper is well written and the diffusion approach in the latent space is an interesting extention to dual autoencoder approaches. With convincing results to support this research.

**Weaknesses:**

My major problem with this manuscript is that the main approach to generating two joint autoencoders is not novel. A similar approach including similar experiments has been proposed and published the approach on dual autoencoder before this submission (https://arxiv.org/pdf/2305.13314) and another publication on dual autoencoder can be found at (https://arxiv.org/pdf/2405.13220). These contributions are neither acknowledged nor cited.  The remaining novelty is the diffusion process within the latent spaces which is by itself an interesting idea and should have been stated as the contribution of this manuscript.

**Questions:**

Please address the weaknesses.

---

> ### Author Response · Authors · 2024-11-19
>
> **Weakness: My major problem with this manuscript is that the main approach to generating two joint autoencoders is not novel. A similar approach including similar experiments has been proposed and published the approach on dual autoencoder before this submission (https://arxiv.org/pdf/2305.13314) and another publication on dual autoencoder can be found at (https://arxiv.org/pdf/2405.13220). These contributions are neither acknowledged nor cited. The remaining novelty is the diffusion process within the latent spaces which is by itself an interesting idea and should have been stated as the contribution of this manuscript.**
>
> Thank you for highlighting this point and bringing these references to our attention. We will ensure both references are cited appropriately in our revised manuscript. However, we would like to clarify that our **primary contribution** lies in the application of a **joint diffusion process** within a shared latent space to refine data pairs in a physically consistent manner, rather than in the use of dual autoencoders.
>
> The dual autoencoder architecture only serves as a preliminary step in constructing the shared latent space, enabling the subsequent diffusion process. To demonstrate the flexibility of our approach, we also tested a modified one-in-two-out autoencoder architecture, where a single encoder and two decoders were employed. This setup maintained the shared latent space, and the joint diffusion process continued to perform effectively, following the discoveries presented in our paper.
>
> Furthermore, the one-in-two-out autoencoder architecture allows for direct application to standard FWI tasks, as demonstrated in our new experiment results (https://bit.ly/4eBIJX2). This experiment highlights the capability of the joint diffusion model to generate accurate velocity maps solely from seismic data input, addressing both novelty and practical application concerns.
>
> To further support our claims, we have provided an example of the generation results using the modified one-in-two-out network architecture at https://bit.ly/4hVLg1q. We hope these additional experiments and clarifications address your concerns regarding the novelty and contributions of our work.

---

> > ### Author Response · Authors · 2024-11-22
> > **Reminder: Follow-up on Rebuttal Responses**
> >
> > Dear Reviewer,
> >
> > We hope this message finds you well. We wanted to kindly remind you that the author-reviewer discussion period is ongoing, and we have posted our responses and revisions over two days ago. Your thoughtful feedback has been immensely valuable in shaping our work, and we greatly appreciate your time and effort in reviewing our manuscript.
> >
> > We have carefully addressed your concerns and answered your questions. If you have any further questions or would like clarification on our responses, we are here and ready to address them promptly. Your input is crucial for refining and improving our manuscript, and we deeply value your insights.
> >
> > We understand your time is valuable, and we sincerely thank you for your dedication to this process. Should you have any additional thoughts or concerns, please do not hesitate to reach out.
> >
> > Thank you again for your attention and support.
> >
> > Best regards,
> > -Authors

---

> > > ### Author Response · Authors · 2024-11-23
> > > **Second Reminder: Follow-up on Rebuttal Responses**
> > >
> > > Dear Reviewer,
> > >
> > > Thank you for your initial feedback on our manuscript. While we did not receive further responses during the discussion period, we have carefully considered your comments and incorporated significant changes to address your concerns. Your observations helped us refine our narrative and ensure that our work emphasizes its core contributions. Thus, we have uploaded a revised version manuscript according to your valuable comments.
> > >
> > > Below are the key revisions specifically addressing your feedback:
> > >
> > > **Section 3.2 (Clarification of the Role of Dual Autoencoder):**
> > >
> > > In response to your feedback, we have revised Section 3.2 to make it clear that the dual autoencoder serves only as a preliminary step in our framework. Its purpose is to establish a shared latent space for the joint diffusion process, but it is not the primary focus or contribution of our work. Additionally, we have cited the suggested references to acknowledge prior work on dual autoencoders and distinguish our approach from these methods.
> > >
> > > **Highlighting the Core Contribution – Joint Diffusion Process:**
> > >
> > > As you noted, the novelty of our work lies in the joint diffusion process. To address this, we have restructured the manuscript to focus on Section 3.3 and the subsequent experiments, which detail how our diffusion process refines seismic-velocity pairs in a shared latent space to adhere to the governing PDE. This approach provides a novel generative perspective on FWI and differentiates our work from others that primarily focus on autoencoder architectures.
> > >
> > > **Strengthened Experimental Analysis:**
> > >
> > > To further demonstrate the capabilities of our method, we have added experiments that showcase the versatility and practicality of the joint diffusion process. This includes a new experiment in Section 4.6, where we demonstrate how the framework performs conventional FWI with only seismic data as input. These results illustrate the model’s ability to address real-world FWI challenges while emphasizing the diffusion process as the core innovation.
> > >
> > > We are confident that these revisions address your concerns and highlight the unique contributions of our work. If there are still areas you feel require improvement, we would greatly appreciate further feedback. Your input has been instrumental in shaping our revisions, and we remain committed to presenting the best version of our research.
> > >
> > > Thank you for your time and consideration.
> > >
> > > Best regards,
> > >
> > > The Authors

---

> > > > ### Author Response · Authors · 2024-11-23
> > > > **Third Reminder: Follow-up on Rebuttal Responses**
> > > >
> > > > Dear Reviewer,
> > > >
> > > > We hope this message finds you well. We wanted to kindly remind you about the author-reviewer discussion period and highlight some important updates we made in response to the feedback received.
> > > >
> > > > In addition to addressing the raised concerns, we conducted a new experiment on noisy seismic data, which **demonstrates the robustness of our *WaveDiffusion* framework under realistic noisy conditions for FWI tasks**. This discovery, though not explicitly requested, adds value by showcasing the method’s versatility and practical applicability. The results of this experiment are summarized in the updated manuscript and can be reviewed in detail via the following links:
> > > >
> > > > The visual comparisons are shown in the figure at https://bit.ly/3V5k0Uc, and the quantitative results are detailed in the table at https://bit.ly/4f0rAqB. The new experiment and results are added to the latest revised manuscript on **Page 19 A. 8 section**.
> > > >
> > > > We would deeply appreciate any feedback you might have on our revisions or this new contribution. Your insights are invaluable to refining our work and ensuring it meets the highest standards.
> > > >
> > > > Thank you again for your time and effort. Please don’t hesitate to reach out if there are any additional concerns or questions regarding the paper.
> > > >
> > > > Best regards,
> > > >
> > > > The authors

---

> > > > > ### Comment · Reviewer_A1pp · 2024-11-24
> > > > >
> > > > > I agree with the other reviewer's assessment of the manuscript, while making an effort to address my concerns, this manuscript requires a more focused approach. The novelty of the diffusion process and inversion strategy should be highlighted prominently. The current emphasis on the auto-encoder dilutes the impact of these key contributions. I believe the paper can be significantly strengthened by shifting its focus towards these core aspects. With these improvements, the paper has the potential to be a valuable contribution to the field. I will maintain my original rating.

---

> > > > > > ### Author Response · Authors · 2024-11-25
> > > > > > **Request for Further Guidance**
> > > > > >
> > > > > > Dear Reviewer A1pp,
> > > > > >
> > > > > > Thank you for your feedback and valuable insights on our manuscript. We greatly appreciate the time and effort you have dedicated to reviewing our work. In the latest revised version (uploaded on Nov. 23), we have carefully addressed each of your comments and concerns. Below, we outline your original comments alongside our corresponding responses, revisions, and additions to the manuscript. We kindly ask if the current revision adequately addresses your concerns, and if not, we would be grateful for your guidance on how to further improve the paper.
> > > > > >
> > > > > >
> > > > > > **Your Original Comment: The main approach to generating two joint autoencoders is not novel.**
> > > > > >
> > > > > > **Our Response:** We fully acknowledge this point and have clarified in **Section 3.2 (page 3-4)** that the dual autoencoder serves as a preliminary step to establish the shared latent space, rather than the primary contribution of our work. To address your concern, we have also cited the suggested references (1, 2) and provided context to differentiate our work.
> > > > > >
> > > > > > **Our Revision to the Manuscript:** The dual autoencoder is now explicitly described as a preliminary mechanism that enables the joint diffusion process. The novelty of our work resides in the joint diffusion process and its ability to refine latent representations within a shared latent space to adhere to physical constraints governed by PDEs.
> > > > > >
> > > > > > This clarification has been added **page 4 line 171-176** in the revised manuscript: "*We emphasize that the dual autoencoder is not the focus of this framework and can be substituted with any architecture capable of producing a combined latent space for seismic data and velocity maps. For instance, the one-encoder-two-decoders architecture in Section 4.6 or an autoencoder incorporating KL divergence could also serve this purpose. The autoencoder primarily facilitates the setup for the joint diffusion process, which is the key innovation in achieving physically consistent seismic-velocity generation.*"
> > > > > >
> > > > > > **Your Original Comment: The diffusion process within the latent spaces should be stated as the main contribution.**
> > > > > >
> > > > > > **Our Response:** We agree with this assessment and have made substantial changes to highlight the novelty of the diffusion process. In Section 3.3 and subsequent sections, we explicitly state that the joint diffusion process is the core contribution of the paper. This process enables the model to trace a path from random initialization to valid solutions governed by the wave equation, offering a new geometric interpretation of FWI.
> > > > > >
> > > > > > **Our Revision to the Manuscript:** We shifted the manuscript's focus to the diffusion process, ensuring that its novelty and impact are clearly conveyed. The **experimental results in Sections 4.3 to 4.6 and Appendix A.2 to A.8** demonstrate the effectiveness and advantages of our joint diffusion framework, including
> > > > > >
> > > > > > *Section 4.3: **Joint diffusion** generation procedure analysis.*
> > > > > >
> > > > > > *Section 4.4: **Joint diffusion** comparison to baseline algorithms (InversionNet).*
> > > > > >
> > > > > > *Section 4.5: **Joint diffusion** comparison to separate diffusion.*
> > > > > >
> > > > > > *Section 4.6: **Joint diffusion** solving a conventional FWI problem and comparison to baseline.*
> > > > > >
> > > > > > *Section A.2: **Joint diffusion** results on more datasets.*
> > > > > >
> > > > > > *Section A.3: **Joint diffusion** results on combined datasets.*
> > > > > >
> > > > > > *Section A.4: **Joint diffusion** results on multiple datasets for data symmetry check and more.*
> > > > > >
> > > > > > *Section A.5: **Joint diffusion** handling noise in seismic data and velocity.*
> > > > > >
> > > > > > *Section A.6: **Joint diffusion** comparison to more baseline algorithms (InversionNet, VelocityGAN, UPFWI).*
> > > > > >
> > > > > > *Section A.7: **Joint diffusion** generated sample usage in data-driven FWI algorithms.*
> > > > > >
> > > > > > *Section A.8: **Joint diffusion** comparison to baseline algorithms (InversionNet, VelocityGAN, UPFWI) on conventional FWI on noisy seismic data input senario.*
> > > > > >
> > > > > > These revisions ensure that the paper's focus is firmly on the joint diffusion process and its unique contributions, minimizing the emphasis on the autoencoder, which serves only as a supporting component.
> > > > > >
> > > > > > **Request for Feedback**
> > > > > >
> > > > > > Given the extensive revisions and new experiments added to address your concerns, we kindly ask if the current manuscript sufficiently addresses the issues you raised. If any aspects remain unresolved, we would be deeply grateful for your feedback on how we can further refine the paper. Your insights are invaluable in ensuring that our work reaches its fullest potential and contributes meaningfully to the field.
> > > > > >
> > > > > > Thank you once again for your time and thoughtful review.
> > > > > >
> > > > > > Best regards,
> > > > > >
> > > > > > The authors

---

> > > > > > > ### Author Response · Authors · 2024-11-26
> > > > > > > **Kind Reminder to Reviewer A1pp**
> > > > > > >
> > > > > > > Dear Reviewer A1pp,
> > > > > > >
> > > > > > > We hope this message finds you well. Thank you once again for your previous comments and feedback on our manuscript. In our last response, we detailed the revisions and additions made to the paper to address the concerns you raised, particularly highlighting the joint diffusion process as the primary contribution and providing extensive supporting experiments and analyses.
> > > > > > >
> > > > > > > We kindly wanted to follow up to ask if our current revision manuscript meet your satisfaction. If you have any further suggestions or specific feedback on the revised manuscript, we would be grateful. Your expertise and insights are invaluable to us, and we remain committed to addressing any remaining issues to improve the clarity, focus, and impact of our work.
> > > > > > >
> > > > > > > If there are particular aspects of the manuscript that you feel could be enhanced further, we would be deeply grateful for your guidance. Your input plays a crucial role in ensuring that our contribution is as robust and meaningful as possible.
> > > > > > >
> > > > > > > Thank you again for your time and dedication to reviewing our work.
> > > > > > >
> > > > > > > Best regards,
> > > > > > >
> > > > > > > The authors

---

> > > > > > > > ### Author Response · Authors · 2024-11-28
> > > > > > > > **Second Reminder to Reviewer A1pp**
> > > > > > > >
> > > > > > > > Dear Reviewer A1pp,
> > > > > > > >
> > > > > > > > We hope this message finds you well. We are writing to kindly follow up on the latest revised version of our manuscript, which was uploaded on November 23. The revisions were made with great care to address your earlier comments, particularly by highlighting the novelty of the joint diffusion process and refining the focus of the manuscript.
> > > > > > > >
> > > > > > > > While we understand your decision to maintain your score, your input on the current version of the manuscript is incredibly valuable to us. If you have any further comments or suggestions, we would be most grateful to receive your guidance on how to enhance the quality and clarity of the paper.
> > > > > > > >
> > > > > > > > Thank you again for your time and thoughtful feedback. We deeply appreciate your engagement and look forward to any additional input you may have.
> > > > > > > >
> > > > > > > > Best regards,
> > > > > > > >
> > > > > > > > The authors

---

> > > > > > > > > ### Author Response · Authors · 2024-12-01
> > > > > > > > > **Third Reminder to Reviewer A1pp: Follow-up on the Revised Manuscript**
> > > > > > > > >
> > > > > > > > > Dear Reviewer A1pp,
> > > > > > > > >
> > > > > > > > > We hope this message finds you well.
> > > > > > > > >
> > > > > > > > > We are reaching out to kindly follow up on the latest revised version of our manuscript, uploaded on November 23. We would greatly appreciate any additional comments or suggestions you may have on the current version.
> > > > > > > > >
> > > > > > > > > Thank you for your time and thoughtful feedback. We look forward to hearing from you.
> > > > > > > > >
> > > > > > > > > Best regards,
> > > > > > > > >
> > > > > > > > > The authors

---

> > > > > > > > > > ### Author Response · Authors · 2024-12-02
> > > > > > > > > > **Fourth Reminder to Reviewer A1pp: Follow-up on the Revised Manuscript**
> > > > > > > > > >
> > > > > > > > > > Dear Reviewer A1pp,
> > > > > > > > > >
> > > > > > > > > > We hope this message finds you well.
> > > > > > > > > >
> > > > > > > > > > We are reaching out to kindly follow up on the latest revised version of our manuscript, uploaded on November 23. We would greatly appreciate any additional comments or suggestions you may have on the current version.
> > > > > > > > > >
> > > > > > > > > > Thank you for your time and thoughtful feedback. We look forward to hearing from you.
> > > > > > > > > >
> > > > > > > > > > Best regards,
> > > > > > > > > >
> > > > > > > > > > The authors

---

> > > > > > > > > > > ### Author Response · Authors · 2024-12-02
> > > > > > > > > > > **Last Day Reminder to Reviewer A1pp: Follow-up on the Revised Manuscript**
> > > > > > > > > > >
> > > > > > > > > > > Dear Reviewer A1pp,
> > > > > > > > > > >
> > > > > > > > > > > We hope this message finds you well.
> > > > > > > > > > >
> > > > > > > > > > > We are reaching out to kindly remind you that today is the **last day** of the discussion session. We would like to kindly ask you for your follow-up on the latest revised version of our manuscript, uploaded on November 23. We would greatly appreciate any additional comments or suggestions you may have on the current version.
> > > > > > > > > > >
> > > > > > > > > > > Thank you for your time and thoughtful feedback. We look forward to hearing from you.
> > > > > > > > > > >
> > > > > > > > > > > Best regards,
> > > > > > > > > > >
> > > > > > > > > > > The authors

---

> > > > > > > > > > > > ### Comment · Reviewer_A1pp · 2024-12-02
> > > > > > > > > > > >
> > > > > > > > > > > > I appreciate your efforts in addressing my concerns. Nevertheless, I will retain my original assessment.

---

### Official Review · Reviewer_1caw · 2024-11-03

**Soundness:** 3
**Presentation:** 3
**Contribution:** 3
**Rating:** 6
**Confidence:** 4

**Summary:**

The manuscript introduces a new approach to invert acoustic wave equation data based on a joint generative process. Although there were earlier papers on the use of generative models for data inversion, the presented approach looks fairly original. The authors study the famous geophysical problem known as full waveform inversion (FWI). The approach was tested on 2D spatial data from public dataset OpenFWI.

**Strengths:**

Generative AI is transforming different industries in our days and its use for data inversion looks like a promising research direction. Both theoretical and experimental parts are well-present and easy-to-follow. An important original feature of the work is joint generation of acoustic data and velocity models.

**Weaknesses:**

1.	It is not clear (at least none of the experiments show this) how to use the presented algorithm to invert actual data. It is shown how to generate acoustic data and velocities. But what is typically expected by the reader is the answer on what to do when we are given with some specific seismic data.
2.	Section 4.2.3 Comparison with Inversionnet is not sufficiently complete and convincing. See Questions below.
3.	The geophysical terminology is mixed in the manuscript. Notice the used wave equation models $acoustic$ data. This is a significant simplification of seismic phenomena. In other words, the terms $acoustic$ and $seismic$ are not interchangeable.

**Questions:**

1.	How acoustic data and velocity models were preprocessed before training?
2.	The authors trained the model for 1000 epochs. How long was it in terms CPU/GPU time (depending on the dataset)?
3.	The discussion in the manuscript covers only generation and inversion of 2D spatial data. While 3D models/data are of much higher interest.  Could the proposed algorithm be used in the 3D case? What will the implication on computational complexity?
4.	An important test for an inversion code is to check that symmetric data with respect to some plane produces a symmetric velocity model. Would the presented generation model obey this principle?
5.	It is not clear from the experiments how the presented algorithm compares to baselines. Section 4.2.3 Comparison with Inversionnet does not give the answer on the obvious question: which of the two algorithms is better.

---

> ### Author Response · Authors · 2024-11-19
> **Addressing Practical Application, Comparisons, and Model Extensions for FWI Using Joint Diffusion**
>
> **Reviewer 1caw**
>
> **Weakness 1: What to do when we are given with some specific seismic data?**
>
> Thank you for highlighting this practical consideration. To address inversion with specific seismic data, we have implemented an adaptable "**one-in-two-out**" configuration within the autoencoder component of our framework. In this setup, only seismic data is input into the encoder, generating a latent representation solely derived from the seismic input. This latent vector is then refined through the joint diffusion process and passed through individual VQ layers and decoders to produce both seismic and velocity outputs.
>
> This approach demonstrates the model's applicability to typical FWI tasks, effectively generating velocity maps from seismic data. Example results and statistical evaluations can be found at http://bit.ly/4eBIJX2 and https://bit.ly/4fyN6DB, respectively. Our joint diffusion model achieves an SSIM of 0.6290, comparable to benchmarks such as InversionNet (SSIM 0.6727) and UPFWI (SSIM 0.6614). While VelocityGAN achieves slightly lower RMSE and MAE, our model is still of the same level in structural accuracy, which is critical for seismic applications.
>
> **Weakness 2: Section 4.2.3 comparing InversionNet lacks detail and context.**
>
> Thank you for your feedback. Section 4.2.3 has been expanded to include detailed context and quantitative comparisons. Results show that WaveDiffusion achieves reasonably same level structural accuracy (SSIM 0.6290) compared to InversionNet and other benchmarks.
>
> **Weakness 3: Geophysical terminology is unclear, and "acoustic" and "seismic" are incorrectly used interchangeably.**
>
> We agree that the terms "acoustic" and "seismic" need clarification. In Section 2, we will explicitly note that "seismic data" refers to "acoustic seismic data" in this study, simplifying wave phenomena for modeling purposes.
>
> **Question 1: How were the seismic data and velocity models preprocessed before training?**
>
> Seismic data and velocity models were resized from [5,70,1000]/[1,70,70] to [3,64,1024]/[1,64,64] (channel, height, depth) for consistency with our architecture. Both were normalized to [-1,1] to ensure compatibility and stability. This will be clarified in the revised manuscript.
>
> **Question 2: How much training time was required in terms of CPU/GPU hours?**
>
> Thank you for your question. Training required approximately 1000 GPU hours for the autoencoder (per dataset) and 2000 GPU hours for the joint diffusion model. This information will be added to the manuscript.
>
> **Question 3: Can the algorithm be extended to 3D data, and what are the computational implications?**
>
> Thank you for this insightful question. Adapting our approach to 3D data is indeed feasible, with some modifications. For instance, the autoencoder module would require a more sophisticated 3D architecture to handle volumetric data, while the latent diffusion process should remain fundamentally similar. A simpler alternative would involve treating the 3D velocity cube as a collection of 2D slices along one horizontal dimension, enabling the use of our current network architecture, like the Appendix M experiment in the OpenFWI paper, though this may reduce volumetric coherence. Computational complexity would increase significantly due to the higher dimensionality. We will add this discussion to the manuscript and explore it as future work.
>
> **Question 4: Does the model preserve symmetry in velocity maps when the input seismic data is symmetric?**
>
> Thank you for this insightful question. To address this, we conducted experiments using the FlatVel_B (FVB) subset, which features symmetric data configurations with pure flat velocity layers. The results, provided at https://bit.ly/3YTjslu, confirm that our WaveDiffusion model successfully preserves symmetry in the generated velocity maps when provided with symmetric seismic data. This demonstrates that the model respects the geometric properties of the input data and maintains consistency in symmetric inversion scenarios. We will include these results and the corresponding discussion in the revised manuscript.
>
> **Question 5: How does the algorithm compare to baselines? Which is better?**
>
> Thank you for your comment. Our primary contribution is not to improve baseline FWI solutions but to introduce a novel generative framework for FWI through the use of joint diffusion models. To the best of our knowledge, there is no prior work that adopts such a generative perspective for FWI, making direct comparisons to baseline methods with similar generative models challenging. Our goal is to provide a new perspective on FWI by exploring its potential as a generative process. However, results from the one-in-two-out experiment (http://bit.ly/4eBIJX2) show that WaveDiffusion achieves an SSIM of 0.6290, reasonably comparable to InversionNet (0.6727) and others in structural accuracy. These findings will be emphasized in the revised manuscript.

---

> > ### Comment · Reviewer_1caw · 2024-11-21
> >
> > Dear authors,
> >
> > I appreciate your responses. My oppinion is that your work is a good fit for Applications to Physical Sciences Area.
> > I'll keep my current score.

---

> > > ### Author Response · Authors · 2024-11-22
> > > **Thank You for Your Feedback**
> > >
> > > Dear Reviewer,
> > >
> > > Thank you for your thoughtful response and for taking the time to engage with our paper. We greatly appreciate your detailed feedback and the effort you’ve put into reviewing our work.
> > >
> > > If there are any additional suggestions or areas where you believe we can further improve the manuscript, we would be most grateful for your guidance.
> > >
> > > Thank you once again for your valuable insights.
> > >
> > > Best regards,
> > > Authors

---

### Author Response · Authors · 2024-11-23
**Submission of Revised Manuscript and Summary of Revisions**

Dear Reviewers,

We hope this message finds you well. We are writing to inform you that we have uploaded the revised version of our manuscript based on your insightful feedback and suggestions. We greatly appreciate the time and effort you invested in reviewing our work and providing constructive comments, which have significantly improved the quality and clarity of our paper. Below is a summary of the main revisions addressing your feedback:

**Section 4.6 (page 9) FWI with Seismic Data Only:**

We added an experiment utilizing the “one-in-two-out” WaveDiffusion model to perform conventional FWI tasks when only seismic data is available. This experiment highlights the model’s capability to invert seismic data into velocity maps effectively.
(***Addressing comments from reviewers 1caw, rGTJ, and STHJ.***)

**Line 687 (page 13) Training Time Details:**
Details about the training times for the autoencoders (1000 GPU hours) and the joint diffusion model (2000 GPU hours) have been added.
(***Addressing reviewer 1caw.***)

**Appendix A.6 (page 16) Comparison with VelocityGAN and UPFWI:**
We conducted additional experiments comparing WaveDiffusion-generated samples with VelocityGAN and UPFWI. This comparison evaluates the effectiveness of our generated samples under different setups.
(***Addressing comments from reviewers 1caw and STHJ.***)

**Section 3.2 (page 4) Paper Novelty Clarification:**
We have cited the suggested references and clarified that the dual autoencoder serves as a preliminary step for implementing the joint diffusion process, rather than being the main contribution of the paper. The primary contribution of our work lies in Section 3.3 (page 4) and is further elaborated upon through the subsequent analysis and experiments. This revision emphasizes the novel aspects of our joint diffusion framework and provides appropriate context for the dual autoencoder's role within the overall methodology.
(***Addressing comments from reviewers rGTJ and A1pp.***)

**Line 105 (page 2) Clarification on Terminology:**
We included a statement clarifying that acoustic seismic data, representing a simplified wave phenomenon, is used in this study.
(***Addressing reviewer 1caw.***)

**Line 688 (page 13) Data Preprocessing Details:**
Details about the data preprocessing steps, including resizing and normalization, have been added for clarity.
(***Addressing reviewer 1caw.***)

**Appendix A.4 (page 14) Symmetry in FVB:**
Joint generation examples across multiple datasets, including FVB, were added to demonstrate the model’s ability to preserve the symmetry of flat-layer structures.
(***Addressing reviewer 1caw.***)

**Appendix A.5 (page 15) Handling Noisy Input Data:**
We included experiments demonstrating the model’s ability to reconstruct clean, noise-free data from noisy inputs, showcasing WaveDiffusion’s robustness under challenging data conditions.
(***Addressing reviewer STHJ.***)

We have carefully revised the manuscript to incorporate your feedback and ensure the improvements are well-documented. If you have any further comments or questions, we would be grateful for your insights. Thank you again for your valuable input and for considering our work.

We look forward to your thoughts on the revised manuscript.

Best regards,

The Authors

---

### Author Response · Authors · 2024-11-23
**Additional Experiment: Handling Noisy Seismic Data**

Dear reviewers,

We would like to share a new discovery by our new experiement that emerged during the course of our revisions: the ***robustness*** of *WaveDiffusion* in ***handling noisy input seismic data in FWI***. While this was not explicitly raised in the initial review, we believe this experiment adds significant value to the paper and further demonstrates the versatility of our method.

In this experiment, we tested *WaveDiffusion*, InversionNet, and VelocityGAN on seismic data corrupted with Gaussian noise (mean=0, std=0.05), a realistic scenario often encountered in field data acquisition. Our results reveal that *WaveDiffusion* robustly refines the noisy seismic data and accurately inverts the corresponding velocity maps, achieving the lowest MAE (0.2227) and RMSE (0.3776) and the highest SSIM (0.6142) among the three methods. Neither InversionNet nor VelocityGAN, which rely on direct image-to-image mapping, were able to handle the noisy inputs effectively.

The visual comparisons are shown in the figure at https://bit.ly/3V5k0Uc, and the quantitative results are detailed in the table at https://bit.ly/4f0rAqB. The new experiment and results are added to the latest revised manuscript on **Page 19 A. 8** section.

Despite being trained only on clean seismic data, *WaveDiffusion* demonstrates its ability to refine noisy latent representations, providing both noise-free seismic data and high-fidelity velocity maps. This capability further emphasizes the strength of the joint diffusion process in real-world scenarios.

We believe this additional evidence underscores the potential of *WaveDiffusion* as a robust and practical tool for FWI tasks, extending its applicability to challenging noisy data conditions.

Best regards,

The Authors

---

### Comment · Reviewer_rGTJ · 2024-11-25
**I stay with my score**

I find the claim that the joint diffusion being the main idea of this paper a bit ridiculous. The idea is discussed from lines 192 to 211, barely a third of a page.
Again, I believe that there maybe novelty in joint diffusion but clearly, the authors did not intend this to be the main contributions originally and the paper was written accordingly.
I understand the pressure to publish at these main conferences, but to be honest, the authors will do more justice to the paper and their own work if they resubmit the paper to the next conference in line. They can revise the paper, be serious about their latest findings (rather than write the referees that they just discovered something major) and write a winning paper that will last.

---

> ### Author Response · Authors · 2024-11-25
>
> Dear Reviewer rGTJ,
>
> Thank you for your latest response. We appreciate your continued engagement and the time you’ve dedicated to reviewing our manuscript. We take your concerns about the framing and emphasis of our contributions seriously and have worked to address them thoroughly in the revised version.
>
> Regarding your comment on the joint diffusion idea being under-discussed in the original manuscript, we would like to note that Section 3.3 is only the steps to perform a joint diffusion process, not the technical analysis part. On the other hand, Section 3.4 provides detailed analysis and technical insights into the joint diffusion process. Additionally, we’ve added extensive experimental results (Sections 4.3–4.6 and Appendix A.2–A.8) to highlight the joint diffusion process as the central contribution of our work.
>
> To ensure that this key contribution is clearly conveyed, we would greatly value your suggestions:
>
> **Experiments:** Is the current experiments sufficient enough to support our joint diffusion part contribution? Are there additional experiments or metrics you would recommend to better validate or expand upon the novelty of the joint diffusion approach?
>
> **Section 3.4 Analysis:** Does the analysis provide sufficient depth to demonstrate the role and impact of joint diffusion? If not, we would be grateful if you could point out areas that need further elaboration or clarification.
>
> We are committed to improving our manuscript and incorporating constructive feedback to ensure it meets the highest standards. Your insights as an expert are invaluable, and we would greatly appreciate your guidance on how to refine the discussion and experiments to fully address your concerns.
>
> Thank you again for your thoughtful comments and for helping us make this a stronger contribution to the field.
>
> Best regards,
>
> The Authors

---

> ### Author Response · Authors · 2024-11-25
> **Response Letter to Reviewer rGTJ about the Main Content in the Manuscript**
>
> Dear Reviewer rGTJ,
>
> Thank you for your continued engagement with our manuscript. We appreciate your comments and have worked diligently to address your concerns regarding the emphasis and presentation of the joint diffusion model, which is indeed the central contribution of our work. Below, we provide a detailed table of contents comparing the joint diffusion content in both the original and revised versions of our manuscript. These demonstrate that the joint diffusion process forms a significant part of the paper.
>
> We kindly request your reconsideration of the manuscript in light of the revisions, expansions, and additional experiments explicitly focusing on the joint diffusion model. If any concerns remain, we would greatly appreciate your specific feedback to further strengthen our work.
>
> ---
>
> ### **Table of Contents Comparison: Joint Diffusion-Related Content**
>
> #### **Original Manuscript**
> 1. **Section 1 (Page 2, Lines 71-96):**
>    General introduction to the joint diffusion model and its significance.
>
> 2. **Section 3.1 (Page 3, Lines 145-157):**
>    Motivation for joint diffusion as a novel approach to refine the shared latent space.
>
> 3. **Section 3.3 (Page 4, Lines 191-210):**
>    Detailed explanation of how the joint diffusion process is implemented on the shared latent space.
>
> 4. **Section 3.4 (Page 4-5, Lines 212-265):**
>    Discoveries and analyses of the joint diffusion model, including:
>    - Refinement of the solution space.
>    - Deviation from the PDE evaluation during forward and backward diffusion processes.
>    - Transformation of the governing PDE process to an SDE.
>
> 5. **Section 4.2.2 (Page 6-8, Lines 312-380):**
>    Joint diffusion results and analysis, including FID scores and noise level relationships.
>
> 6. **Section 4.2.3 (Page 8, Lines 381-431):**
>    Using joint diffusion-generated samples to train InversionNet.
>
> 7. **Section 4.2.4 (Page 9, Lines 432-469):**
>    Comparison between joint diffusion and separate diffusion for seismic and velocity data.
>
> 8. **Appendix A.1.2 (Page 13, Lines 665-677):**
>    Hyperparameter details for the joint diffusion model.
>
> 9. **Appendix A.2 (Page 13, Lines 686-697):**
>    Joint diffusion results on FVB and FFB datasets.
>
> 10. **Appendix A.3 (Page 13, Lines 686-697):**
>     Joint diffusion results on combined datasets.
>
> **Figures and Tables in Original Manuscript:**
> - **Figure 1:** Overview of WaveDiffusion and the joint diffusion process.
> - **Figure 4:** Joint diffusion architecture.
> - **Figures 5–7:** Analysis of joint diffusion results and deviation from the PDE.
> - **Table 1–4:** FID scores, comparisons with baselines, and separate vs. joint diffusion results.
>
> ---
>
> #### **Revised Manuscript**
> To address your concerns and further emphasize the joint diffusion process, we added the following new content:
>
> 1. **Section 4.6 (Page 9, Lines 436-457):**
>    Application of the joint diffusion model to conventional FWI problems with seismic data only.
>
> 2. **Appendix A.4 (Page 14-15, Lines 743-789):**
>    Joint diffusion-generated samples visualization and data symmetry checks for flat velocity layers.
>
> 3. **Appendix A.5 (Page 15-16, Lines 791-844):**
>    Joint diffusion handling noisy seismic and velocity data, demonstrating robust reconstruction.
>
> 4. **Appendix A.6 (Page 16-18, Lines 846-960):**
>    FWI performance comparisons between multiple baseline algorithms and joint diffusion.
>
> 5. **Appendix A.8 (Page 19, Lines 991-1023):**
>    Joint diffusion model performance on noisy seismic data input, highlighting its robustness and applicability.
>
> **New Figures and Tables in Revised Manuscript:**
> - **Figure 10:** Visualization of FWI results with joint diffusion.
> - **Figure 14:** Joint diffusion-generated examples across multiple subsets.
> - **Figure 16:** Reconstructions of noisy seismic and velocity data using joint diffusion.
> - **Figure 18:** FWI results visualization with noisy input seismic data.
> - **Tables 5, 6, and 8:** Comparative metrics for joint diffusion vs. baselines.
>
> ---
>
> ### Key Revisions and Justifications
>
> 1. **Expanded Technical Description (Sections 3.3, 3.4):**
>    Detailed implementation of joint diffusion, including:
>    - Refinement of latent space through joint diffusion.
>    - Evaluation of deviations from the governing PDE during diffusion processes.
>    - Transforming PDE processes into SDEs for physical consistency.
>
> 2. **Comprehensive Experiments (Sections 4.2.2–4.6, Appendices A.4–A.8):**
>    Extensive results showcasing the effectiveness of joint diffusion:
>    - Robust handling of noisy seismic inputs (Appendix A.5).
>    - Symmetry checks and visualizations across multiple datasets (Appendix A.4).
>    - Comparisons with multiple baseline methods, including InversionNet, VelocityGAN, and UPFWI (Appendices A.6, A.8).
>
> ---
>
>
> Thank you for your time and effort in reviewing our work.
>
> Best regards,
> The authors

---

> ### Comment · Reviewer_A1pp · 2024-11-26
>
> I here agree with reviewer rGTJ that while joint diffusion may offer novel applications, it seems this wasn't the primary focus of the author's original intent. I will stay with my score too.

---

> > ### Author Response · Authors · 2024-11-26
> > **Follow-Up to Reviewer A1pp**
> >
> > Dear Reviewer A1pp,
> >
> > Thank you for your feedback and for sharing your perspective. We greatly appreciate the time and effort you have devoted to reviewing our manuscript. While we understand your decision to maintain your score, we would like to kindly ask if you have any comments or suggestions regarding the latest revised version of the manuscript.
> >
> > Your expertise is invaluable to us, and we are committed to improving the quality and clarity of our work. If there are specific aspects of the revised manuscript that you believe could be further refined or strengthened, we would be deeply grateful for your insights. Your guidance will help us ensure that the manuscript meets the highest standards of scientific rigor and presentation.
> >
> > Thank you again for your thoughtful engagement throughout this process. We look forward to hearing any additional thoughts you may have.
> >
> > Best regards,
> >
> > The authors

---

> ### Author Response · Authors · 2024-11-27
> **Reminder to Reviewer rGTJ for Follow-Up**
>
> Dear Reviewer rGTJ,
>
> Thank you for your feedback and engagement with our manuscript. We deeply value the time and effort you have spent reviewing our work. While we understand your decision to maintain your score, we would like to kindly ask for any further comments or suggestions you may have regarding the **latest revised version of the manuscript**.
>
> We have worked diligently to address the concerns raised in your earlier comments and made significant revisions to clarify and emphasize the novelty of the joint diffusion process, its applications, and its analysis. Your expertise is invaluable to us, and we are committed to further improving the manuscript based on your insights.
>
> If there are specific aspects of the revised manuscript that you believe could benefit from additional refinement or focus, we would greatly appreciate your guidance. Your feedback will help us ensure the manuscript is of the highest quality and effectively communicates its contributions to the field.
>
> Thank you again for your thoughtful review and input. We look forward to hearing any additional suggestions you may have.
>
> Best regards,
> The authors

---

### Meta-Review · Area_Chair_HaVE · 2024-12-21

**Metareview:**

This paper addresses full waveform inversion by training autoencoders on models and data, with shared latent spaces, followed by training a diffusion model in the latent space to generate samples. While the reviewers acknowledged the idea as reasonable and potentially effective, the contribution and presentation fall significantly below ICLR standards.

Two reviewers noted that paired autoencoders have been applied to inverse problems before, and multimodal generative models have long explored similar ideas (some of which the authors acknowledge). Unfortunately, these reviewers did not substantively engage in the discussion (please see below), which rightly frustrated the authors, but the core critique remains valid.

The authors emphasize joint diffusion as the primary contribution, but this is effectively just latent diffusion applied to an highly stylized problem (layered phantoms with few degrees of freedom). The two channels are generated by two decoders whose semantics are entirely determined by the first stage. Whether or not paired autoencoders are considered part of the contributio, the method relies on them—or equivalent architectures—in such a way that the remainder is not novel enough for publication. Once a joint latent space is learned, various generative models could be employed.

Another key issue is the lack of discussion on the tradeoffs between conditional and joint generation. There are numerous involved tradeoffs which are not mentioned, let alone investigated through experiments. The paper’s claim of tackling the harder problem of simultaneous generation across modalities is valid, but it is important to know why? What are the tradeoffs? How do they manifest? What are the right experiments or possible theoretical insights? Joint generative modeling is generally harder than discriminative, where observations localize or guide the generator, especially if there is a cheap approximate inverses.

The problem of stability, key for inverse problems, is unaddressed. It would be informative to train a model with noisy measurements (at varying noise levels), which is different from inputting noisy data into a trained model. The current joint distribution is degenerate—measurements are functions of unknowns and we can sample joint dist by generating models and training an FNO.The authors report thousands of GPU hours to train their models on simple phantoms, with massive network architectures, whereas much smaller nets have achieved strong results in similar settings. The justification for this computational expense is unclear.

The paper’s presentation is another issue. Design choices are not clearly motivated or lack explanation. The VQ is a crucial part of the model. The authors say it ensures a "compact and discrete representation," but what does this mean? Why does VQ enable the model to "capture structured patterns and long-range dependencies"? Also, what exactly is being vector-quantized? The supplementary mentions 8192 32-dimensional vectors—is that for the entire 16×16 latent? If it is then in Section 4.6 the reconstruction consists in picking one out of 8192 templates. What is really the case remains unclear.

Same is the case for the latent-space diffusion ("joint diffusion"). It is described so tersely and handwavily that it is unclear how the model is trained; the supplementary contains information about hyperparameters and learning rates but not about inputs, outputs, losses, ... . The only equation in the first block is z_t = z + eps_t (which is missing a _t), but in the second block t seems to be the terminal time, L is never defined so we don't know what it is, ..., there is little chance to reproduce results.

Experiments are also problematic. The method is tested only on simple layered phantoms (very few DoFs) with an acoustic wave equation, comparing to a couple of deep learning baselines (Table 5). Other experiments analyze the proposed method, without establishing its utility. The authors claim they are not focused on achieving strong reconstruction results, but no clear alternative insights emerge. The experiments with separate AEs show that learning the joint distribution of synthetic data is simpler than solving the ill-posed full waveform inversion. It would be interesting to navigate this spectrum. The paper's repeated references to "seismic data" are misleading, as it uses simulated 2D acoustic data from 64x64 layered toy models. Real-world geophysical models are far large, 3D, and waves are elastic. While academic compute limits are understood (although this does not seem to be a block here), the extensive resources for such stylized setups are hard to justify.

**Additional Comments On Reviewer Discussion:**

rGTJ and A1pp were both critical, identifying novelty as the key issue. STHJ and 1caw thought the paper is a borderline accept. During discussion the authors reiterated that the main focus is on "joint diffusion" rather than the paired autoencoders. rGTJ argued that "clearly, the authors did not intend this to be the main contributions originally and the paper was written accordingly" by pointing out that only a handful are dedicated to joint diffusion, and A1pp concurred. I do not agree with this since joint diffusion has been in the title from the get-go, not autoencoders. But I do agree about the real estate dedicated to joint diffusion. The problem is that all ideas in the paper, not just this one, are presented in a telegraphic way and it often reads a bit like a sequence of arbitrary choices.

The authors are justified in pointing out that rGTJ and A1pp were mostly disengaged from the discussion. That is suboptimal, but the issue is that it is hard to make up for the missing novelty by running new experiments. I believe that the paper and the experiments need a major overhaul.

My recommendation is based on a careful analysis of all the responses and my personal very detailed reading of the paper and familiarity with both the hardcore inverse problems and SciML literature.

---

### Decision · Program_Chairs · 2025-01-22

Reject